# A Nonconvex Optimization Framework for Low Rank Matrix Estimation*

**Tuo Zhao**
Johns Hopkins University

**Zhaoran Wang**      **Han Liu**
Princeton University

## Abstract

We study the estimation of low rank matrices via nonconvex optimization. Compared with convex relaxation, nonconvex optimization exhibits superior empirical performance for large scale instances of low rank matrix estimation. However, the understanding of its theoretical guarantees are limited. In this paper, we define the notion of projected oracle divergence based on which we establish sufficient conditions for the success of nonconvex optimization. We illustrate the consequences of this general framework for matrix sensing. In particular, we prove that a broad class of nonconvex optimization algorithms, including alternating minimization and gradient-type methods, geometrically converge to the global optimum and exactly recover the true low rank matrices under standard conditions.

## 1   Introduction

Let $M^* \in \mathbb{R}^{m \times n}$ be a rank $k$ matrix with $k$ much smaller than $m$ and $n$. Our goal is to estimate $M^*$ based on partial observations of its entires. For example, matrix sensing is based on linear measurements $\langle A_i, M^* \rangle$, where $i \in \{1, \ldots, d\}$ with $d$ much smaller than $mn$ and $A_i$ is the sensing matrix. In the past decade, significant progress has been established on the recovery of low rank matrix [4, 5, 23, 18, 15, 16, 12, 22, 7, 25, 19, 6, 14, 11, 13, 8, 9, 10, 27]. Among all these existing works, most are based upon convex relaxation with nuclear norm constraint or regularization. Nevertheless, solving these convex optimization problems can be computationally prohibitive in high dimensional regimes with large $m$ and $n$ [27]. A computationally more efficient alternative is nonconvex optimization. In particular, we reparameterize the $m \times n$ matrix variable $M$ in the optimization problem as $UV^\top$ with $U \in \mathbb{R}^{m \times k}$ and $V \in \mathbb{R}^{n \times k}$, and optimize over $U$ and $V$. Such a reparametrization automatically enforces the low rank structure and leads to low computational cost per iteration. Due to this reason, the nonconvex approach is widely used in large scale applications such as recommendation systems [17].

Despite the superior empirical performance of the nonconvex approach, the understanding of its theoretical guarantees is relatively limited in comparison with the convex relaxation approach. Only until recently has there been progress on coordinate descent-type nonconvex optimization methods, which is known as alternating minimization [14, 8, 9, 10]. They show that, provided a desired initialization, the alternating minimization algorithm converges at a geometric rate to $U^* \in \mathbb{R}^{m \times k}$ and $V^* \in \mathbb{R}^{n \times k}$, which satisfy $M = U^*V^{*\top}$. Meanwhile, [15, 16] establish the convergence of gradient-type methods, and [27] further establish the convergence of a broad class of nonconvex algorithms including both gradient-type and coordinate descent-type methods. However, [15, 16, 27] only establish the asymptotic convergence for an infinite number of iterations, rather than the explicit rate of convergence. Besides these works, [18, 12, 13] consider projected gradient-type methods, which optimize over the matrix variable $M \in \mathbb{R}^{m \times n}$ rather than $U \in \mathbb{R}^{m \times k}$ and $V \in \mathbb{R}^{n \times k}$. These methods involve calculating the top $k$ singular vectors of an $m \times n$ matrix at each iteration. For

$k$ much smaller than $m$ and $n$, they incur much higher computational cost per iteration than the aforementioned methods that optimize over $U$ and $V$. All these works, except [27], focus on specific algorithms, while [27] do not establish the explicit optimization rate of convergence.

In this paper, we propose a general framework that unifies a broad class of nonconvex algorithms for low rank matrix estimation. At the core of this framework is a quantity named projected oracle divergence, which sharply captures the evolution of generic optimization algorithms in the presence of nonconvexity. Based on the projected oracle divergence, we establish sufficiently conditions under which the iteration sequences geometrically converge to the global optima. For matrix sensing, a direct consequence of this general framework is that, a broad family of nonconvex algorithms, including gradient descent, coordinate gradient descent and coordinate descent, converge at a geometric rate to the true low rank matrices $U^*$ and $V^*$. In particular, our general framework covers alternating minimization as a special case and recovers the results of [14, 8, 9, 10] under standard conditions. Meanwhile, our framework covers gradient-type methods, which are also widely used in practice [28, 24]. To the best of our knowledge, our framework is the first one that establishes exact recovery guarantees and geometric rates of convergence for a broad family of nonconvex matrix sensing algorithms.

To achieve maximum generality, our unified analytic framework significantly differs from previous works. In detail, [14, 8, 9, 10] view alternating minimization as a perturbed version of the power method. However, their point of view relies on the closed form solution of each iteration of alternating minimization, which makes it hard to generalize to other algorithms, e.g., gradient-type methods. Meanwhile, [27] take a geometric point of view. In detail, they show that the global optimum of the optimization problem is the unique stationary point within its neighborhood and thus a broad class of algorithms succeed. However, such geometric analysis of the objective function does not characterize the convergence rate of specific algorithms towards the stationary point. Unlike existing analytic frameworks, we analyze nonconvex optimization algorithms as perturbed versions of their convex counterparts. For example, under our framework we view alternating minimization as a perturbed version of coordinate descent on convex objective functions. We use the key quantity, projected oracle divergence, to characterize such a perturbation effect, which results from the local nonconvexity at intermediate solutions. This framework allows us to establish explicit rate of convergence in an analogous way as existing convex optimization analysis.

**Notation:** For a vector $v = (v_1, \ldots, v_d)^T \in \mathbb{R}^d$, let the vector $\ell_q$ norm be $\|v\|_q^q = \sum_j v_j^q$. For a matrix $A \in \mathbb{R}^{m \times n}$, we use $A_{*j} = (A_{1j}, ..., A_{mj})^\top$ to denote the $j$-th column of $A$, and $A_{i*} = (A_{i1}, ..., A_{in})^\top$ to denote the $i$-th row of $A$. Let $\sigma_{\max}(A)$ and $\sigma_{\min}(A)$ be the largest and smallest nonzero singular values of $A$. We define the following matrix norms: $\|A\|_{\mathrm{F}}^2 = \sum_j \|A_{*j}\|_2^2$, $\|A\|_2 = \sigma_{\max}(A)$. Moreover, we define $\|A\|_*$ to be the sum of all singular values of $A$. Given another matrix $B \in \mathbb{R}^{m \times n}$, we define the inner product as $\langle A, B \rangle = \sum_{i,j} A_{ij} B_{ij}$. We define $e_i$ as an indicator vector, where the $i$-th entry is one, and all other entries are zero. For a bivariate function $f(u, v)$, we define $\nabla_u f(u, v)$ to be the gradient with respect to $u$. Moreover, we use the common notations of $\Omega(\cdot)$, $O(\cdot)$, and $o(\cdot)$ to characterize the asymptotics of two real sequences.

## 2   Problem Formulation and Algorithms

Let $M^* \in \mathbb{R}^{m \times n}$ be the unknown low rank matrix of interest. We have $d$ sensing matrices $A_i \in \mathbb{R}^{m \times n}$ with $i \in \{1, \ldots, d\}$. Our goal is to estimate $M^*$ based on $b_i = \langle A_i, M^* \rangle$ in the high dimensional regime with $d$ much smaller than $mn$. Under such a regime, a common assumption is $\mathrm{rank}(M^*) = k \ll \min\{d, m, n\}$. Existing approaches generally recover $M^*$ by solving the following convex optimization problem

$$\min_{M \in \mathbb{R}^{m \times n}} \|M\|_* \quad \text{subject to } b = \mathcal{A}(M), \tag{2.1}$$

where $b = [b_1, ..., b_d]^\top \in \mathbb{R}^d$, and $\mathcal{A}(M) : \mathbb{R}^{m \times n} \to \mathbb{R}^d$ is an operator defined as

$$\mathcal{A}(M) = [\langle A_1, M \rangle, ..., \langle A_i, M \rangle]^\top \in \mathbb{R}^d. \tag{2.2}$$

Existing convex optimization algorithms for solving (2.1) are computationally inefficient, in the sense that they incur high per-iteration computational cost, and only attain sublinear rates of convergence to the global optimum [14]. Instead, in large scale settings we usually consider the following nonconvex

optimization problem

$$\min_{U\in\mathbb{R}^{m\times k},V\in\mathbb{R}^{n\times k}}\mathcal{F}(U,V). \quad \text{where } \mathcal{F}(U,V)=\frac{1}{2}\|b-\mathcal{A}(UV^{\top})\|_2^2. \tag{2.3}$$

The reparametrization of $M = UV^{\top}$, though making the optimization problem in (2.3) nonconvex, significantly improves the computational efficiency. Existing literature [17, 28, 21, 24] has established convincing empirical evidence that (2.3) can be effectively solved by a board variety of gradient-based nonconvex optimization algorithms, including gradient descent, alternating exact minimization (i.e., alternating least squares or coordinate descent), as well as alternating gradient descent (i.e., coordinate gradient descent), which are shown in Algorithm 1.

It is worth noting the QR decomposition and rank $k$ singular value decomposition in Algorithm 1 can be accomplished efficiently. In particular, the QR decomposition can be accomplished in $O(k^2 \max\{m,n\})$ operations, while the rank $k$ singular value decomposition can be accomplished in $O(kmn)$ operations. In fact, the QR decomposition is not necessary for particular update schemes, e.g., [14] prove that the alternating exact minimization update schemes with or without the QR decomposition are equivalent.

---

**Algorithm 1** A family of nonconvex optimization algorithms for matrix sensing. Here $(\overline{U}, D, \overline{V}) \leftarrow$ KSVD$(M)$ is the rank $k$ singular value decomposition of $M$. Here $D$ is a diagonal matrix containing the top $k$ singular values of $M$ in decreasing order, and $\overline{U}$ and $\overline{V}$ contain the corresponding top $k$ left and right singular vectors of $M$. Here $(\overline{V}, R_{\overline{V}}) \leftarrow$ QR$(V)$ is the QR decomposition, where $\overline{V}$ is the corresponding orthonormal matrix and $R_{\overline{V}}$ is the corresponding upper triangular matrix.

---

**Input**: $\{b_i\}_{i=1}^d, \{A_i\}_{i=1}^d$
**Parameter**: Step size $\eta$, Total number of iterations $T$
$(\overline{U}^{(0)}, D^{(0)}, \overline{V}^{(0)}) \leftarrow$ KSVD$(\sum_{i=1}^d b_i A_i)$, $V^{(0)} \leftarrow \overline{V}^{(0)} D^{(0)}$, $U^{(0)} \leftarrow \overline{U}^{(0)} D^{(0)}$
**For:** $t = 0, \ldots., T-1$

    Alternating Exact Minimization : $V^{(t+0.5)} \leftarrow \operatorname{argmin}_V \mathcal{F}(\overline{U}^{(t)}, V)$
    $(\overline{V}^{(t+1)}, R_{\overline{V}}^{(t+0.5)}) \leftarrow$ QR$(V^{(t+0.5)})$
    Alternating Gradient Descent : $V^{(t+0.5)} \leftarrow V^{(t)} - \eta \nabla_V \mathcal{F}(\overline{U}^{(t)}, V^{(t)})$
    $(\overline{V}^{(t+1)}, R_{\overline{V}}^{(t+0.5)}) \leftarrow$ QR$(V^{(t+0.5)})$, $U^{(t)} \leftarrow \overline{U}^{(t)} R_{\overline{V}}^{(t+0.5)\top}$
    Gradient Descent : $V^{(t+0.5)} \leftarrow V^{(t)} - \eta \nabla_V \mathcal{F}(\overline{U}^{(t)}, V^{(t)})$
    $(\overline{V}^{(t+1)}, R_{\overline{V}}^{(t+0.5)}) \leftarrow$ QR$(V^{(t+0.5)})$, $U^{(t+1)} \leftarrow \overline{U}^{(t)} R_{\overline{V}}^{(t+0.5)\top}$    } Updating $V$

    Alternating Exact Minimization : $U^{(t+0.5)} \leftarrow \operatorname{argmin}_U \mathcal{F}(U, \overline{V}^{(t+1)})$
    $(\overline{U}^{(t+1)}, R_{\overline{U}}^{(t+0.5)}) \leftarrow$ QR$(U^{(t+0.5)})$
    Alternating Gradient Descent : $U^{(t+0.5)} \leftarrow U^{(t)} - \eta \nabla_U \mathcal{F}(U^{(t)}, \overline{V}^{(t+1)})$
    $(\overline{U}^{(t+1)}, R_{\overline{U}}^{(t+0.5)}) \leftarrow$ QR$(U^{(t+0.5)})$, $V^{(t+1)} \leftarrow \overline{V}^{t+1} R_{\overline{U}}^{(t+0.5)\top}$    } Updating $U$
    Gradient Descent : $U^{(t+0.5)} \leftarrow U^{(t)} - \eta \nabla_U \mathcal{F}(U^{(t)}, \overline{V}^{(t)})$
    $(\overline{U}^{(t+1)}, R_{\overline{U}}^{(t+0.5)}) \leftarrow$ QR$(U^{(t+0.5)})$, $V^{(t+1)} \leftarrow \overline{V}^{t} R_{\overline{U}}^{(t+0.5)\top}$
**End for**
**Output:** $M^{(T)} \leftarrow U^{(T-0.5)} \overline{V}^{(T)\top}$ (for gradient descent we use $\overline{U}^{(T)} V^{(T)\top}$)

---

## 3 Theoretical Analysis

We analyze the convergence properties of the general family of nonconvex optimization algorithms illustrated in §2. Before we present the main results, we first introduce a unified analytic framework based on a key quantity named projected oracle divergence. Such a unified framework equips our theory with the maximum generality. Without loss of generality, we assume $m \le n$ throughout the rest of this paper.

### 3.1 Projected Oracle Divergence

We first provide an intuitive explanation for the success of nonconvex optimization algorithms, which forms the basis of our later proof for the main results. Recall that (2.3) is a special instance of the following optimization problem,

$$\min_{U\in\mathbb{R}^{m\times k},V\in\mathbb{R}^{n\times k}} f(U,V). \tag{3.1}$$

A key observation is that, given fixed $U$, $f(U, \cdot)$ is strongly convex and smooth in $V$ under suitable conditions, and the same also holds for $U$ given fixed $V$ correspondingly. For the convenience of

discussion, we summarize this observation in the following technical condition, which will be later verified for matrix sensing under suitable conditions.

**Condition 3.1** (Strong Biconvexity and Bismoothness). There exist universal constants $\mu_+ > 0$ and $\mu_- > 0$ such that

$$\frac{\mu_-}{2}\|U' - U\|_{\mathrm{F}}^2 \leq f(U', V) - f(U, V) - \langle U' - U, \nabla_U f(U, V) \rangle \leq \frac{\mu_+}{2}\|U' - U\|_{\mathrm{F}}^2 \text{ for all } U, U',$$

$$\frac{\mu_-}{2}\|V' - V\|_{\mathrm{F}}^2 \leq f(U, V') - f(U, V) - \langle V' - V, \nabla_V f(U, V) \rangle \leq \frac{\mu_+}{2}\|V' - V\|_{\mathrm{F}}^2 \text{ for all } V, V'.$$

For the simplicity of discussion, for now we assume $U^*$ and $V^*$ are the unique global minimizers to the generic optimization problem in (3.1). Assuming $U^*$ is given, we can obtain $V^*$ by

$$V^* = \underset{V \in \mathbb{R}^{n \times k}}{\operatorname{argmin}} f(U^*, V). \tag{3.2}$$

Condition 3.1 implies the objective function in (3.2) is strongly convex and smooth. Hence, we can choose any gradient-based algorithm to obtain $V^*$. For example, we can directly solve for $V^*$ in

$$\nabla_V f(U^*, V) = 0, \tag{3.3}$$

or iteratively solve for $V^*$ using gradient descent, i.e.,

$$V^{(t)} = V^{(t-1)} - \eta \nabla_V f(U^*, V^{(t-1)}), \tag{3.4}$$

where $\eta$ is the step size. For the simplicity of discussion, we put aside the renormalization issue for now. In the example of gradient descent, by invoking classical convex optimization results [20], it is easy to prove that

$$\|V^{(t)} - V^*\|_{\mathrm{F}} \leq \kappa \|V^{(t-1)} - V^*\|_{\mathrm{F}} \text{ for all } t = 0, 1, 2, \ldots,$$

where $\kappa \in (0, 1)$ is a contraction coefficient, which depends on $\mu_+$ and $\mu_-$ in Condition 3.1. However, the first-order oracle $\nabla_V f(U^*, V^{(t-1)})$ is not accessible in practice, since we do not know $U^*$. Instead, we only have access to $\nabla_V f(U, V^{(t-1)})$, where $U$ is arbitrary. To characterize the divergence between the ideal first-order oracle $\nabla_V f(U^*, V^{(t-1)})$ and the accessible first-order oracle $\nabla_V f(U, V^{(t-1)})$, we define a key quantity named projected oracle divergence, which takes the form

$$\mathcal{D}(V, V', U) = \langle \nabla_V f(U^*, V') - \nabla_V f(U, V'), V - V^*/(\|V - V^*\|_{\mathrm{F}}) \rangle, \tag{3.5}$$

where $V'$ is the point for evaluating the gradient. In the above example, it holds for $V' = V^{(t-1)}$. Later we will illustrate that, the projection of the difference of first-order oracles onto a specific one dimensional space, i.e., the direction of $V - V^*$, is critical to our analysis. In the above example of gradient descent, we will prove later that for $V^{(t)} = V^{(t-1)} - \eta \nabla_V f(U, V^{(t-1)})$, we have

$$\|V^{(t)} - V^*\|_{\mathrm{F}} \leq \kappa \|V^{(t-1)} - V^*\|_{\mathrm{F}} + 2/\mu_+ \cdot \mathcal{D}(V^{(t)}, V^{(t-1)}, U). \tag{3.6}$$

In other words, the projection of the divergence of first-order oracles onto the direction of $V^{(t)} - V^*$ captures the perturbation effect of employing the accessible first-order oracle $\nabla_V f(U, V^{(t-1)})$ instead of the ideal $\nabla_V f(U^*, V^{(t-1)})$. For $V^{(t+1)} = \operatorname{argmin}_V f(U, V)$, we will prove that

$$\|V^{(t)} - V^*\|_{\mathrm{F}} \leq 1/\mu_- \cdot \mathcal{D}(V^{(t)}, V^{(t)}, U). \tag{3.7}$$

According to the update schemes shown in Algorithm 1, for alternating exact minimization, we set $U = U^{(t)}$ in (3.7), while for gradient descent or alternating gradient descent, we set $U = U^{(t-1)}$ or $U = U^{(t)}$ in (3.6) respectively. Correspondingly, similar results hold for $\|U^{(t)} - U^*\|_{\mathrm{F}}$.

To establish the geometric rate of convergence towards the global minima $U^*$ and $V^*$, it remains to establish upper bounds for the projected oracle divergence. In the example of gradient decent we will prove that for some $\alpha \in (0, 1 - \kappa)$,

$$2/\mu_+ \cdot \mathcal{D}(V^{(t)}, V^{(t-1)}, U^{(t-1)}) \leq \alpha \|U^{(t-1)} - U^*\|_{\mathrm{F}},$$

which together with (3.6) (where we take $U = U^{(t-1)}$) implies

$$\|V^{(t)} - V^*\|_{\mathrm{F}} \leq \kappa \|V^{(t-1)} - V^*\|_{\mathrm{F}} + \alpha \|U^{(t-1)} - U^*\|_{\mathrm{F}}. \tag{3.8}$$

Correspondingly, similar results hold for $\|U^{(t)} - U^*\|_{\mathrm{F}}$, i.e.,

$$\|U^{(t)} - U^*\|_{\mathrm{F}} \leq \kappa \|U^{(t-1)} - U^*\|_{\mathrm{F}} + \alpha \|V^{(t-1)} - V^*\|_{\mathrm{F}}. \tag{3.9}$$

Combining (3.8) and (3.9) we then establish the contraction

$$\max\{\|V^{(t)} - V^*\|_{\mathrm{F}}, \|U^{(t)} - U^*\|_{\mathrm{F}}\} \leq (\alpha + \kappa) \cdot \max\{\|V^{(t-1)} - V^*\|_{\mathrm{F}}, \|U^{(t-1)} - U^*\|_{\mathrm{F}}\},$$

which further implies the geometric convergence, since $\alpha \in (0, 1 - \kappa)$. Respectively, we can establish similar results for alternating exact minimization and alternating gradient descent. Based upon such a unified analytic framework, we now simultaneously establish the main results.

**Remark 3.2.** Our proposed projected oracle divergence is inspired by previous work [3, 2, 1], which analyzes the Wirtinger Flow algorithm for phase retrieval, the expectation maximization (EM) Algorithm for latent variable models, and the gradient descent algorithm for sparse coding. Though their analysis exploits similar nonconvex structures, they work on completely different problems, and the delivered technical results are also fundamentally different.

## 3.2 Matrix Sensing

Before we present our main results, we first introduce an assumption known as the restricted isometry property (RIP). Recall that $k$ is the rank of the target low rank matrix $M^*$.

**Assumption 3.3.** The linear operator $\mathcal{A}(\cdot) : \mathbb{R}^{m \times n} \to \mathbb{R}^d$ defined in (2.2) satisfies $2k$-RIP with parameter $\delta_{2k} \in (0, 1)$, i.e., for all $\Delta \in \mathbb{R}^{m \times n}$ such that $\text{rank}(\Delta) \leq 2k$, it holds that

$$(1 - \delta_{2k})\|\Delta\|_{\mathrm{F}}^2 \leq \|\mathcal{A}(\Delta)\|_2^2 \leq (1 + \delta_{2k})\|\Delta\|_{\mathrm{F}}^2.$$

Several random matrix ensembles satisfy $k$-RIP for a sufficiently large $d$ with high probability. For example, suppose that each entry of $A_i$ is independently drawn from a sub-Gaussian distribution, $\mathcal{A}(\cdot)$ satisfies $2k$-RIP with parameter $\delta_{2k}$ with high probability for $d = \Omega(\delta_{2k}^{-2} kn \log n)$.

The following theorem establishes the geometric rate of convergence of the nonconvex optimization algorithms summarized in Algorithm 1.

**Theorem 3.4.** Assume there exists a sufficiently small constant $C_1$ such that $\mathcal{A}(\cdot)$ satisfies $2k$-RIP with $\delta_{2k} \leq C_1/k$, and the largest and smallest nonzero singular values of $M^*$ are constants, which do not scale with $(d, m, n, k)$. For any pre-specified precision $\epsilon$, there exist an $\eta$ and universal constants $C_2$ and $C_3$ such that for all $T \geq C_2 \log(C_3/\epsilon)$, we have $\|M^{(T)} - M^*\|_{\mathrm{F}} \leq \epsilon$.

The proof of Theorems 3.4 is provided in Appendices 4.1, A.1, and A.2. Theorem 3.4 implies that all three nonconvex optimization algorithms geometrically converge to the global optimum. Moreover, assuming that each entry of $A_i$ is independently drawn from a sub-Gaussian distribution with mean zero and variance proxy one, our result further suggests, to achieve exact low rank matrix recovery, our algorithm requires the number of measurements $d$ to satisfy

$$d = \Omega(k^3 n \log n), \tag{3.10}$$

since we assume that $\delta_{2k} \leq C_1/k$. This sample complexity result matches the state-of-the-art result for nonconvex optimization methods, which is established by [14]. In comparison with their result, which only covers the alternating exact minimization algorithm, our results holds for a broader variety of nonconvex optimization algorithms.

Note that the sample complexity in (3.10) depends on a polynomial of $\sigma_{\max}(M^*)/\sigma_{\min}(M^*)$, which is treated as a constant in our paper. If we allow $\sigma_{\max}(M^*)/\sigma_{\min}(M^*)$ to increase with the dimension, we can plug the nonconvex optimization algorithms into the multi-stage framework proposed by [14]. Following similar lines to the proof of Theorem 3.4, we can derive a new sample complexity, which is independent of $\sigma_{\max}(M^*)/\sigma_{\min}(M^*)$. See more details in [14].

# 4 Proof of Main Results

Due to space limitation, we only sketch the proof of Theorem 3.4 for alternating exact minimization. The proof of Theorem 3.4 for alternating gradient descent and gradient descent, and related lemmas are provided in the appendix. For notational simplicity, let $\sigma_1 = \sigma_{\max}(M^*)$ and $\sigma_k = \sigma_{\min}(M^*)$. Before we proceed with the main proof, we first introduce the following lemma, which verifies Condition 3.1.

**Lemma 4.1.** Suppose that $\mathcal{A}(\cdot)$ satisfies $2k$-RIP with parameter $\delta_{2k}$. Given an arbitrary orthonormal matrix $\overline{U} \in \mathbb{R}^{m \times k}$, for any $V$, $V' \in \mathbb{R}^{n \times k}$, we have

$$\frac{1 + \delta_{2k}}{2}\|V' - V\|_{\mathrm{F}}^2 \geq \mathcal{F}(\overline{U}, V') - \mathcal{F}(\overline{U}, V) - \langle \nabla_V \mathcal{F}(\overline{U}, V), V' - V \rangle \geq \frac{1 - \delta_{2k}}{2}\|V' - V\|_{\mathrm{F}}^2.$$

The proof of Lemma 4.1 is provided in Appendix B.1. Lemma 4.1 implies that $\mathcal{F}(\overline{U}, \cdot)$ is strongly convex and smooth in $V$ given a fixed orthonormal matrix $\overline{U}$, as specified in Condition 3.1. Equipped with Lemma 4.1, we now lay out the proof for each update scheme in Algorithm 1.

## 4.1 Proof of Theorem 3.4 (Alternating Exact Minimization)

*Proof.* Throughout the proof of alternating exact minimization, we define a constant $\xi \in (1, \infty)$ for notational simplicity. We assume that at the $t$-th iteration, there exists a matrix factorization of $M^* = \overline{U}^{*(t)} V^{*(t)\top}$, where $\overline{U}^{*(t)}$ is orthonormal. We choose the projected oracle divergence as

$$\mathcal{D}(V^{(t+0.5)}, V^{(t+0.5)}, \overline{U}^{(t)}) = \left\langle \nabla_V \mathcal{F}(\overline{U}^{*(t)}, V^{(t+0.5)}) - \nabla_V \mathcal{F}(\overline{U}^{(t)}, V^{(t+0.5)}), \frac{V^{(t+0.5)} - V^{*(t)}}{\|V^{(t+0.5)} - V^{*(t)}\|_{\mathrm{F}}} \right\rangle.$$

**Remark 4.2.** Note that the matrix factorization is not necessarily unique. Because given a factorization of $M^* = UV^\top$, we can always obtain a new factorization of $M^* = \widetilde{U}\widetilde{V}^\top$, where $\widetilde{U} = UO$ and $\widetilde{V} = VO$ for an arbitrary unitary matrix $O \in \mathbb{R}^{k \times k}$. However, this is not a issue to our convergence analysis. As will be shown later, we can prove that there always exists a factorization of $M^*$ satisfying the desired computational properties for each iteration (See Lemma 4.5, Corollaries 4.7 and 4.8).

The following lemma establishes an upper bound for the projected oracle divergence.

**Lemma 4.3.** Suppose that $\delta_{2k}$ and $\overline{U}^{(t)}$ satisfy

$$\delta_{2k} \leq \frac{\sqrt{2}(1 - \delta_{2k})^2 \sigma_k}{4\xi k(1 + \delta_{2k})\sigma_1} \quad \text{and} \quad \|\overline{U}^{(t)} - \overline{U}^{*(t)}\|_{\mathrm{F}} \leq \frac{(1 - \delta_{2k})\sigma_k}{4\xi(1 + \delta_{2k})\sigma_1}. \tag{4.1}$$

Then we have $\mathcal{D}(V^{(t+0.5)}, V^{(t+0.5)}, \overline{U}^{(t)}) \leq \frac{(1 - \delta_{2k})\sigma_k}{2\xi}\|\overline{U}^{(t)} - \overline{U}^{*(t)}\|_{\mathrm{F}}$.

The proof of Lemma 4.3 is provided in Appendix B.2. Lemma 4.3 shows that the projected oracle divergence for updating $V$ diminishes with the estimation error of $\overline{U}^{(t)}$. The following lemma quantifies the progress of an exact minimization step using the projected oracle divergence.

**Lemma 4.4.** We have $\|V^{(t+0.5)} - V^{*(t)}\|_{\mathrm{F}} \leq 1/(1 - \delta_{2k}) \cdot \mathcal{D}(V^{(t+0.5)}, V^{(t+0.5)}, \overline{U}^{(t)})$.

The proof of Lemma 4.4 is provided in Appendix B.3. Lemma 4.4 illustrates that the estimation error of $V^{(t+0.5)}$ diminishes with the projected oracle divergence. The following lemma characterizes the effect of the renormalization step using QR decomposition.

**Lemma 4.5.** Suppose that $V^{(t+0.5)}$ satisfies

$$\|V^{(t+0.5)} - V^{*(t)}\|_{\mathrm{F}} \leq \sigma_k/4. \tag{4.2}$$

Then there exists a factorization of $M^* = U^{*(t+1)}\overline{V}^{*(t+1)}$ such that $\overline{V}^{*(t+0.5)} \in \mathbb{R}^{n \times k}$ is an orthonormal matrix, and satisfies $\|\overline{V}^{(t+1)} - \overline{V}^{*(t+1)}\|_{\mathrm{F}} \leq 2/\sigma_k \cdot \|V^{(t+0.5)} - V^{*(t)}\|_{\mathrm{F}}$.

The proof of Lemma 4.5 is provided in Appendix B.4. The next lemma quantifies the accuracy of the initialization $\overline{U}^{(0)}$.

**Lemma 4.6.** Suppose that $\delta_{2k}$ satisfies

$$\delta_{2k} \leq \frac{(1 - \delta_{2k})^2 \sigma_k^4}{192\xi^2 k(1 + \delta_{2k})^2 \sigma_1^4}. \tag{4.3}$$

Then there exists a factorization of $M^* = \overline{U}^{*(0)} V^{*(0)\top}$ such that $\overline{U}^{*(0)} \in \mathbb{R}^{m \times k}$ is an orthonormal matrix, and satisfies $\|\overline{U}^{(0)} - \overline{U}^*\|_{\mathrm{F}} \leq \frac{(1 - \delta_{2k})\sigma_k}{4\xi(1 + \delta_{2k})\sigma_1}$.

The proof of Lemma 4.6 is provided in Appendix B.5. Lemma 4.6 implies that the initial solution $\overline{U}^{(0)}$ attains a sufficiently small estimation error.

Combining the above Lemmas, we obtain the next corollary for a complete iteration of updating $V$.

**Corollary 4.7.** Suppose that $\delta_{2k}$ and $\overline{U}^{(t)}$ satisfy

$$\delta_{2k} \leq \frac{(1 - \delta_{2k})^2 \sigma_k^4}{192\xi^2 k(1 + \delta_{2k})^2 \sigma_1^4} \quad \text{and} \quad \|\overline{U}^{(t)} - \overline{U}^{*(t)}\|_{\mathrm{F}} \leq \frac{(1 - \delta_{2k})\sigma_k}{4\xi(1 + \delta_{2k})\sigma_1}. \tag{4.4}$$

We then have $\|\overline{V}^{(t+1)} - \overline{V}^{*(t+1)}\|_{\mathrm{F}} \leq \frac{(1 - \delta_{2k})\sigma_k}{4\xi(1 + \delta_{2k})\sigma_1}$. Moreover, we also have $\|\overline{V}^{(t+1)} - \overline{V}^{*(t+1)}\|_{\mathrm{F}} \leq \frac{1}{\xi}\|\overline{U}^{(t)} - \overline{U}^{*(t)}\|_{\mathrm{F}}$ and $\|V^{(t+0.5)} - V^{*(t)}\|_{\mathrm{F}} \leq \frac{\sigma_k}{2\xi}\|\overline{U}^{(t)} - \overline{U}^{*(t)}\|_{\mathrm{F}}$.

The proof of Corollary 4.7 is provided in Appendix B.6. Since the alternating exact minimization algorithm updates $U$ and $V$ in a symmetric manner, we can establish similar results for a complete iteration of updating $U$ in the next corollary.

**Corollary 4.8.** Suppose that $\delta_{2k}$ and $\overline{V}^{(t+1)}$ satisfy

$$\delta_{2k} \leq \frac{(1-\delta_{2k})^2 \sigma_k^4}{192\xi^2 k(1+\delta_{2k})^2 \sigma_1^4} \quad \text{and} \quad \|\overline{V}^{(t+1)} - \overline{V}^{*(t+1)}\|_{\mathrm{F}} \leq \frac{(1-\delta_{2k})\sigma_k}{4\xi(1+\delta_{2k})\sigma_1}. \qquad (4.5)$$

Then there exists a factorization of $M^* = \overline{U}^{*(t+1)} V^{*(t+1)\top}$ such $\overline{U}^{*(t+1)}$ is an orthonormal matrix, and satisfies $\|\overline{U}^{(t+1)} - \overline{U}^{*(t+1)}\|_{\mathrm{F}} \leq \frac{(1-\delta_{2k})\sigma_k}{4\xi(1+\delta_{2k})\sigma_1}$. Moreover, we also have $\|\overline{U}^{(t+1)} - \overline{U}^{*(t+1)}\|_{\mathrm{F}} \leq \frac{1}{\xi}\|\overline{V}^{(t+1)} - \overline{V}^{*(t+1)}\|_{\mathrm{F}}$ and $\|U^{(t+0.5)} - U^{*(t+1)}\|_{\mathrm{F}} \leq \frac{\sigma_k}{2\xi}\|\overline{V}^{(t+1)} - \overline{V}^{*(t+1)}\|_{\mathrm{F}}$.

The proof of Corollary 4.8 directly follows Appendix B.6, and is therefore omitted.

We then proceed with the proof of Theorem 3.4 for alternating exact minimization. Lemma 4.6 ensures that (4.4) of Corollary 4.7 holds for $\overline{U}^{(0)}$. Then Corollary 4.7 ensures that (4.5) of Corollary 4.8 holds for $\overline{V}^{(1)}$. By induction, Corollaries 4.7 and 4.8 can be applied recursively for all $T$ iterations. Thus we obtain

$$\|\overline{V}^{(T)} - \overline{V}^{*(T)}\|_{\mathrm{F}} \leq \frac{1}{\xi}\|\overline{U}^{(T-1)} - \overline{U}^{*(T-1)}\|_{\mathrm{F}} \leq \frac{1}{\xi^2}\|\overline{V}^{(T-1)} - \overline{V}^{*(T-1)}\|_{\mathrm{F}}$$

$$\leq \cdots \leq \frac{1}{\xi^{2T-1}}\|\overline{U}^{(0)} - \overline{U}^{*(0)}\|_{\mathrm{F}} \leq \frac{(1-\delta_{2k})\sigma_k}{4\xi^{2T}(1+\delta_{2k})\sigma_1}, \qquad (4.6)$$

where the last inequality comes from Lemma 4.6. Therefore, for a pre-specified accuracy $\epsilon$, we need at most $T = \left\lceil 1/2 \cdot \log\left(\frac{(1-\delta_{2k})\sigma_k}{2\epsilon(1+\delta_{2k})\sigma_1}\right) \log^{-1}\xi \right\rceil$ iterations such that

$$\|\overline{V}^{(T)} - \overline{V}^{*(T)}\|_{\mathrm{F}} \leq \frac{(1-\delta_{2k})\sigma_k}{4\xi^{2T}(1+\delta_{2k})\sigma_1} \leq \frac{\epsilon}{2}. \qquad (4.7)$$

Moreover, Corollary 4.8 implies

$$\|U^{(T-0.5)} - U^{*(T)}\|_{\mathrm{F}} \leq \frac{\sigma_k}{2\xi}\|\overline{V}^{(T)} - \overline{V}^{*(T)}\|_{\mathrm{F}} \leq \frac{(1-\delta_{2k})\sigma_k^2}{8\xi^{2T+1}(1+\delta_{2k})\sigma_1},$$

where the last inequality comes from (4.6). Therefore, we need at most

$$T = \left\lceil 1/2 \cdot \log\left(\frac{(1-\delta_{2k})\sigma_k^2}{4\xi\epsilon(1+\delta_{2k})}\right) \log^{-1}\xi \right\rceil$$

iterations such that

$$\|U^{(T-0.5)} - U^*\|_{\mathrm{F}} \leq \frac{(1-\delta_{2k})\sigma_k^2}{8\xi^{2T+1}(1+\delta_{2k})\sigma_1} \leq \frac{\epsilon}{2\sigma_1}. \qquad (4.8)$$

Then combining (4.7) and (4.8), we obtain

$$\|M^{(T)} - M^*\| = \|U^{(T-0.5)}\overline{V}^{(T)\top} - U^{*(T)}\overline{V}^{*(T)\top}\|_{\mathrm{F}}$$

$$\leq \|\overline{V}^{(T)}\|_2\|U^{(T-0.5)} - U^{*(T)}\|_{\mathrm{F}} + \|U^{*(T)}\|_2\|\overline{V}^{(T)} - \overline{V}^{*(T)}\|_{\mathrm{F}} \leq \epsilon, \qquad (4.9)$$

where the last inequality is from $\|\overline{V}^{(T)}\|_2 = 1$ (since $\overline{V}^{(T)}$ is orthonormal) and $\|U^*\|_2 = \|M^*\|_2 = \sigma_1$ (since $U^{*(T)}\overline{V}^{*(T)\top} = M^*$ and $\overline{V}^{*(T)}$ is orthonormal). Thus we complete the proof. $\qquad \square$

# 5 Extension to Matrix Completion

Under the same setting as matrix sensing, we observe a subset of the entries of $M^*$, namely, $\mathcal{W} \subseteq \{1, \ldots, m\} \times \{1, \ldots, n\}$. We assume that $\mathcal{W}$ is drawn uniformly at random, i.e., $M^*_{i,j}$ is observed independently with probability $\bar{\rho} \in (0, 1]$. To exactly recover $M^*$, a common assumption is the incoherence of $M^*$, which will be specified later. A popular approach for recovering $M^*$ is to solve the following convex optimization problem

$$\min_{M \in \mathbb{R}^{m \times n}} \|M\|_* \quad \text{subject to } \mathcal{P}_{\mathcal{W}}(M^*) = \mathcal{P}_{\mathcal{W}}(M), \qquad (5.1)$$

where $\mathcal{P}_{\mathcal{W}}(M) : \mathbb{R}^{m \times n} \to \mathbb{R}^{m \times n}$ is an operator defined as $[\mathcal{P}_{\mathcal{W}}(M)]_{ij} = M_{ij}$ if $(i, j) \in \mathcal{W}$, and $0$ otherwise. Similar to matrix sensing, existing algorithms for solving (5.1) are computationally

inefficient. Hence, in practice we usually consider the following nonconvex optimization problem

$$\min_{U\in\mathbb{R}^{m\times k},V\in\mathbb{R}^{n\times k}} \mathcal{F}_{\mathcal{W}}(U,V), \quad \text{where } \mathcal{F}_{\mathcal{W}}(U,V) = 1/2\cdot\|\mathcal{P}_{\mathcal{W}}(M^*) - \mathcal{P}_{\mathcal{W}}(UV^\top)\|_{\text{F}}^2. \quad (5.2)$$

Similar to matrix sensing, (5.2) can also be efficiently solved by gradient-based algorithms. Due to space limitation, we present these matrix completion algorithms in Algorithm 2 of Appendix D. For the convenience of later convergence analysis, we partition the observation set $\mathcal{W}$ into $2T+1$ subsets $\mathcal{W}_0,...,\mathcal{W}_{2T}$ using Algorithm 4 in Appendix D. However, in practice we do not need the partition scheme, i.e., we simply set $\mathcal{W}_0 = \cdots = \mathcal{W}_{2T} = \mathcal{W}$.

Before we present the main results, we introduce an assumption known as the incoherence property.

**Assumption 5.1.** The target rank $k$ matrix $M^*$ is incoherent with parameter $\mu$, i.e., given the rank $k$ singular value decomposition of $M^* = \overline{U}^*\Sigma^*\overline{V}^{*\top}$, we have

$$\max_i \|\overline{U}^*_{i*}\|_2 \leq \mu\sqrt{k/m} \quad \text{and} \quad \max_j \|\overline{V}^*_{j*}\|_2 \leq \mu\sqrt{k/n}.$$

The incoherence assumption guarantees that $M^*$ is far from a sparse matrix, which makes it feasible to complete $M^*$ when its entries are missing uniformly at random. The following theorem establishes the iteration complexity and the estimation error under the Frobenius norm.

**Theorem 5.2.** Suppose that there exists a universal constant $C_4$ such that $\bar{\rho}$ satisfies

$$\bar{\rho} \geq C_4\mu^2 k^3 \log n \log(1/\epsilon)/m, \quad (5.3)$$

where $\epsilon$ is the pre-specified precision. Then there exist an $\eta$ and universal constants $C_5$ and $C_6$ such that for any $T \geq C_5 \log(C_6/\epsilon)$, we have $\|M^{(T)} - M\|_{\text{F}} \leq \epsilon$ with high probability.

Due to space limit, we defer the proof of Theorem 5.2 to the longer version of this paper. Theorem 5.2 implies that all three nonconvex optimization algorithms converge to the global optimum at a geometric rate. Furthermore, our results indicate that the completion of the true low rank matrix $M^*$ up to $\epsilon$-accuracy requires the entry observation probability $\bar{\rho}$ to satisfy

$$\bar{\rho} = \Omega(\mu^2 k^3 \log n \log(1/\epsilon)/m). \quad (5.4)$$

This result matches the result established by [8], which is the state-of-the-art result for alternating minimization. Moreover, our analysis covers three nonconvex optimization algorithms.

# 6 Experiments

We present numerical experiments for matrix sensing to support our theoretical analysis. We choose $m = 30$, $n = 40$, and $k = 5$, and vary $d$ from 300 to 900. Each entry of $A_i$'s are independent sampled from $N(0,1)$. We then generate $M = UV^\top$, where $\widetilde{U} \in \mathbb{R}^{m\times k}$ and $\widetilde{V} \in \mathbb{R}^{n\times k}$ are two matrices with all their entries independently sampled from $N(0,1/k)$. We then generate $d$ measurements by $b_i = \langle A_i, M \rangle$ for $i = 1,...,d$. Figure 1 illustrates the empirical performance of the alternating exact minimization and alternating gradient descent algorithms for a single realization. The step size for the alternating gradient descent algorithm is determined by the backtracking line search procedure. We see that both algorithms attain linear rate of convergence for $d = 600$ and $d = 900$. Both algorithms fail for $d = 300$, because $d = 300$ is below the minimum requirement of sample complexity for the exact matrix recovery.

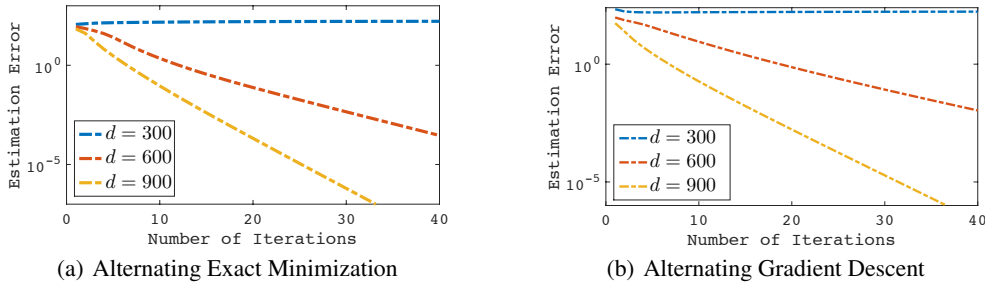

(a) Alternating Exact Minimization        (b) Alternating Gradient Descent

Figure 1: Two illustrative examples for matrix sensing. The vertical axis corresponds to estimation error $\|M^{(t)} - M\|_{\text{F}}$. The horizontal axis corresponds to numbers of iterations. Both the alternating exact minimization and alternating gradient descent algorithms attain linear rate of convergence for $d = 600$ and $d = 900$. But both algorithms fail for $d = 300$, because $d = 300$ is below the minimum requirement of sample complexity for the exact matrix recovery.

## Footnotes

*Research supported by NSF IIS1116730, NSF IIS1332109, NSF IIS1408910, NSF IIS1546482-BIGDATA, NSF DMS1454377-CAREER, NIH R01GM083084, NIH R01HG06841, NIH R01MH102339, and FDA HHSF223201000072C.

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
