[Supplementary Material]

# A Proof of Main Results for Matrix Sensing (Cont'd)

We continue with the proof of Theorem 3.4 for the alternating gradient and gradient descent algorithms.

## A.1 Proof of Theorem 3.4 (Alternating Gradient Descent)

*Proof.* Throughout the proof for alternating gradient descent, we define a sufficiently large constant $\xi$. Moreover, we assume that at the $t$-th iteration, there exists a matrix factorization of $M^*$

$$M^* = \overline{U}^{*(t)} V^{*(t)\top},$$

where $\overline{U}^{*(t)} \in \mathbb{R}^{m \times k}$ is an orthonormal matrix. We use the following projected oracle divergence

$$\mathcal{D}(V^{(t+0.5)}, V^{(t)}, \overline{U}^{(t)}) = \left\langle \nabla_V \mathcal{F}(\overline{U}^{(t)}, V^{(t)}) - \nabla_V \mathcal{F}(\overline{U}^{*(t)}, V^{(t)}), \frac{V^{(t+0.5)} - V^{*(t)}}{\|V^{(t+0.5)} - V^{*(t)}\|_{\mathrm{F}}} \right\rangle.$$

The first lemma is parallel to Lemma 4.3 for alternating exact minimization.

**Lemma A.1.** Suppose that $\delta_{2k}, \overline{U}^{(t)}$, and $V^{(t)}$ satisfy

$$\delta_{2k} \leq \frac{\sqrt{2}(1-\delta_{2k})\sigma_k}{8\xi k \sigma_1}, \quad \|\overline{U}^{(t)} - \overline{U}^{*(t)}\|_{\mathrm{F}} \leq \frac{\sigma_k^2}{4\xi\sigma_1^2}, \quad \text{and} \quad \|V^{(t)} - V^{*(t)}\|_{\mathrm{F}} \leq \frac{\sigma_1\sqrt{k}}{2}. \quad (\text{A.1})$$

Then we have

$$\mathcal{D}(V^{(t+0.5)}, V^{(t)}, \overline{U}^{(t)}) \leq \frac{(1+\delta_{2k})\sigma_k}{\xi} \|\overline{U}^{(t)} - \overline{U}^{*(t)}\|_{\mathrm{F}}.$$

The proof of Lemma A.1 is provided in Appendix C.1. Lemma A.1 illustrates the projected oracle divergence diminishes with the estimation error of $\overline{U}^{(t)}$, when $\overline{U}^{(t)}$ and $V^{(t)}$ are sufficiently close to $\overline{U}^{*(t)}$ and $V^{*(t)}$.

**Lemma A.2.** Suppose that the step size parameter $\eta$ satisfies

$$\eta = \frac{1}{1+\delta_{2k}}. \quad (\text{A.2})$$

Then we have

$$\|V^{(t+0.5)} - V^*\|_{\mathrm{F}} \leq \sqrt{\delta_{2k}} \|V^{(t)} - V^*\|_{\mathrm{F}} + \frac{2}{1+\delta_{2k}} \mathcal{D}(V^{(t+0.5)}, V^{(t)}, \overline{U}^{(t)}).$$

The proof of Lemma A.2 is in Appendix C.2. Lemma A.2 characterizes the progress of a gradient descent step with a pre-specified fixed step size. A more practical option is adaptively selecting $\eta$ using the backtracking line search procedure, and similar results can be guaranteed. See [20] for details. The following lemma characterizes the effect of the renormalization step using QR decomposition.

**Lemma A.3.** Suppose that $V^{(t+0.5)}$ satisfies

$$\|V^{(t+0.5)} - V^{*(t)}\|_{\mathrm{F}} \leq \frac{\sigma_k}{4}. \quad (\text{A.3})$$

Then there exists a factorization of $M^* = U^{*(t+1)} \overline{V}^{*(t+1)}$ such that $\overline{V}^{*(t+1)} \in \mathbb{R}^{n \times k}$ is an orthonormal matrix, and

$$\|\overline{V}^{(t+1)} - \overline{V}^{*(t+1)}\|_{\mathrm{F}} \leq \frac{2}{\sigma_k} \|V^{(t+0.5)} - V^{*(t)}\|_{\mathrm{F}},$$

$$\|U^{(t)} - U^{*(t+1)}\|_{\mathrm{F}} \leq \frac{3\sigma_1}{\sigma_k} \|V^{(t+0.5)} - V^{*(t)}\|_{\mathrm{F}} + \sigma_1 \|\overline{U}^{(t)} - \overline{U}^{*(t)}\|_{\mathrm{F}},$$

The proof of Lemma A.3 is provided in Appendix C.3. The next lemma quantifies the accuracy of the initial solutions.

**Lemma A.4.** Suppose that $\delta_{2k}$ satisfies

$$\delta_{2k} \leq \frac{\sigma_k^6}{192\xi^2 k \sigma_1^6}. \quad (\text{A.4})$$

Then we have

$$\|\overline{U}^{(0)} - \overline{U}^{*(0)}\|_{\mathrm{F}} \leq \frac{\sigma_k^2}{4\xi\sigma_1^2} \quad \text{and} \quad \|V^{(0)} - V^{*(0)}\|_{\mathrm{F}} \leq \frac{\sigma_k^2}{2\xi\sigma_1} \leq \frac{\sigma_1\sqrt{k}}{2}.$$

The proof of Lemma A.4 is in Appendix C.4. Lemma A.4 indicates that the initial solutions $\overline{U}^{(0)}$ and $V^{(0)}$ attain sufficiently small estimation errors.

Combining Lemmas A.1, A.2, 4.5, , we obtain the following corollary for a complete iteration of updating $V$.

**Corollary A.5.** Suppose that $\delta_{2k}$, $\overline{U}^{(t)}$, and $V^{(t)}$ satisfy

$$\delta_{2k} \leq \frac{\sigma_k^6}{192\xi^2 k\sigma_1^6}, \quad \|\overline{U}^{(t)} - \overline{U}^{*(t)}\|_{\mathrm{F}} \leq \frac{\sigma_k^2}{4\xi\sigma_1^2}, \quad \text{and} \quad \|V^{(t)} - V^{*(t)}\|_{\mathrm{F}} \leq \frac{\sigma_k^2}{2\xi\sigma_1}. \qquad (A.5)$$

We then have

$$\|\overline{V}^{(t+1)} - \overline{V}^{*(t+1)}\|_{\mathrm{F}} \leq \frac{\sigma_k^2}{4\xi\sigma_1^2} \quad \text{and} \quad \|U^{(t)} - U^{*(t+1)}\|_{\mathrm{F}} \leq \frac{\sigma_k^2}{2\xi\sigma_1}.$$

Moreover, we have

$$\|V^{(t+0.5)} - V^{*(t)}\|_{\mathrm{F}} \leq \sqrt{\delta_{2k}}\|V^{(t)} - V^{*(t)}\|_{\mathrm{F}} + \frac{2\sigma_k}{\xi}\|\overline{U}^{(t)} - \overline{U}^{*(t)}\|_{\mathrm{F}}, \qquad (A.6)$$

$$\|\overline{V}^{(t+1)} - \overline{V}^{*(t+1)}\|_{\mathrm{F}} \leq \frac{2\sqrt{\delta_{2k}}}{\sigma_k}\|V^{(t)} - V^{*(t)}\|_{\mathrm{F}} + \frac{4}{\xi}\|\overline{U}^{(t)} - \overline{U}^{*(t)}\|_{\mathrm{F}}, \qquad (A.7)$$

$$\|U^{(t)} - U^{*(t+1)}\|_{\mathrm{F}} \leq \frac{3\sigma_1\sqrt{\delta_{2k}}}{\sigma_k}\|V^{(t)} - V^{*(t)}\|_{\mathrm{F}} + \left(\frac{6}{\xi} + 1\right)\sigma_1\|\overline{U}^{(t)} - \overline{U}^{*(t)}\|_{\mathrm{F}}. \qquad (A.8)$$

The proof of Corollary A.5 is provided in Appendix C.5. Since the alternating gradient descent algorithm updates $U$ and $V$ in a symmetric manner, we can establish similar results for a complete iteration of updating $U$ in the next corollary.

**Corollary A.6.** Suppose that $\delta_{2k}$, $\overline{V}^{(t+1)}$, and $U^{(t)}$ satisfy

$$\delta_{2k} \leq \frac{\sigma_k^6}{192\xi^2 k\sigma_1^6}, \quad \|\overline{V}^{(t+1)} - \overline{V}^{*(t+1)}\|_{\mathrm{F}} \leq \frac{\sigma_k^2}{4\xi\sigma_1^2}, \quad \text{and} \quad \|U^{(t)} - U^{*(t+1)}\|_{\mathrm{F}} \leq \frac{\sigma_k^2}{2\xi\sigma_1}. \qquad (A.9)$$

We then have

$$\|\overline{U}^{(t+1)} - \overline{U}^{*(t+1)}\|_{\mathrm{F}} \leq \frac{\sigma_k^2}{4\xi\sigma_1^2} \quad \text{and} \quad \|V^{(t+1)} - V^{*(t+1)}\|_{\mathrm{F}} \leq \frac{\sigma_k^2}{2\xi\sigma_1}.$$

Moreover, we have

$$\|U^{(t+0.5)} - U^{*(t+1)}\|_{\mathrm{F}} \leq \sqrt{\delta_{2k}}\|U^{(t)} - U^{*(t+1)}\|_{\mathrm{F}} + \frac{2\sigma_k}{\xi}\|\overline{V}^{(t+1)} - \overline{V}^{*(t+1)}\|_{\mathrm{F}}, \qquad (A.10)$$

$$\|\overline{U}^{(t+1)} - \overline{U}^{*(t+1)}\|_{\mathrm{F}} \leq \frac{2\sqrt{\delta_{2k}}}{\sigma_k}\|U^{(t)} - U^{*(t+1)}\|_{\mathrm{F}} + \frac{4}{\xi}\|\overline{V}^{(t+1)} - \overline{V}^{*(t+1)}\|_{\mathrm{F}}, \qquad (A.11)$$

$$\|V^{(t+1)} - V^{*(t+1)}\|_{\mathrm{F}} \leq \frac{3\sigma_1\sqrt{\delta_{2k}}}{\sigma_k}\|U^{(t)} - U^{*(t+1)}\|_{\mathrm{F}} + \left(\frac{6}{\xi} + 1\right)\sigma_1\|\overline{V}^{(t+1)} - \overline{V}^{*(t+1)}\|_{\mathrm{F}}.$$
$$(A.12)$$

The proof of Corollary A.6 directly follows Appendix C.5, and is therefore omitted..

Now we proceed with the proof of Theorem 3.4 for alternating gradient descent. Recall that Lemma A.4 ensures that (A.5) of Corollary A.5 holds for $\overline{U}^{(0)}$ and $V^{(0)}$. Then Corollary A.5 ensures that (A.9) of Corollary A.6 holds for $U^{(0)}$ and $\overline{V}^{(1)}$. By induction, Corollaries 4.7 and 4.8 can be applied recursively for all $T$ iterations. For notational simplicity, we write (A.6)-(A.12) as

$$\|V^{(t+0.5)} - V^{*(t)}\|_{\mathrm{F}} \leq \alpha_1\|V^{(t)} - V^{*(t)}\|_{\mathrm{F}} + \gamma_1\sigma_1\|\overline{U}^{(t)} - \overline{U}^{*(t)}\|_{\mathrm{F}}, \qquad (A.13)$$

$$\sigma_1\|\overline{V}^{(t+1)} - \overline{V}^{*(t+1)}\|_{\mathrm{F}} \leq \alpha_2\|V^{(t)} - V^{*(t)}\|_{\mathrm{F}} + \gamma_2\sigma_1\|\overline{U}^{(t)} - \overline{U}^{*(t)}\|_{\mathrm{F}}, \qquad (A.14)$$

$$\|U^{(t+0.5)} - U^{*(t+1)}\|_{\mathrm{F}} \leq \alpha_3\|U^{(t)} - U^{*(t+1)}\|_{\mathrm{F}} + \gamma_3\sigma_1\|\overline{V}^{(t+1)} - \overline{V}^{*(t+1)}\|_{\mathrm{F}}, \qquad (A.15)$$

$$\sigma_1\|\overline{U}^{(t+1)} - \overline{U}^{*(t+1)}\|_{\mathrm{F}} \leq \alpha_4\|U^{(t)} - U^{*(t+1)}\|_{\mathrm{F}} + \gamma_4\sigma_1\|\overline{V}^{(t+1)} - \overline{V}^{*(t+1)}\|_{\mathrm{F}}, \qquad (A.16)$$

$$\|U^{(t)} - U^{*(t+1)}\|_{\mathrm{F}} \leq \alpha_5\|V^{(t)} - V^{*(t)}\|_{\mathrm{F}} + \gamma_5\sigma_1\|\overline{U}^{(t)} - \overline{U}^{*(t)}\|_{\mathrm{F}}, \qquad (A.17)$$

$$\|V^{(t+1)} - V^{*(t+1)}\|_{\mathrm{F}} \leq \alpha_6\|U^{(t)} - U^{*(t+1)}\|_{\mathrm{F}} + \gamma_6\sigma_1\|\overline{V}^{(t+1)} - \overline{V}^{*(t+1)}\|_{\mathrm{F}}. \qquad (A.18)$$

Note that we have $\gamma_5, \gamma_6 \in (1,2)$, but $\alpha_1,...,\alpha_6$, $\gamma_1,...,$ and $\gamma_4$ can be sufficiently small as long as $\xi$ is sufficiently large. We then have

$$\|U^{(t+1)} - U^{*(t+2)}\|_{\mathrm{F}} \overset{(i)}{\leq} \alpha_5\|V^{(t+1)} - V^{*(t+1)}\|_{\mathrm{F}} + \gamma_5\sigma_1\|\overline{U}^{(t+1)} - \overline{U}^{*(t+1)}\|_{\mathrm{F}}$$

$$\overset{(ii)}{\leq} \alpha_5\alpha_6\|U^{(t)} - U^{*(t+1)}\|_{\mathrm{F}} + \alpha_5\gamma_6\sigma_1\|\overline{V}^{(t+1)} - \overline{V}^{*(t+1)}\|_{\mathrm{F}} + \gamma_5\sigma_1\|\overline{U}^{(t+1)} - \overline{U}^{*(t+1)}\|_{\mathrm{F}}$$

$$\overset{(iii)}{\leq} (\alpha_5\alpha_6 + \gamma_5\alpha_4)\|U^{(t)} - U^{*(t+1)}\|_{\mathrm{F}} + (\gamma_5\gamma_4\sigma_1 + \alpha_5\gamma_6)\sigma_1\|\overline{V}^{(t+1)} - \overline{V}^{*(t+1)}\|_{\mathrm{F}}$$

$$\overset{(iv)}{\leq} (\alpha_5\alpha_6 + \gamma_5\alpha_4)\|U^{(t)} - U^{*(t+1)}\|_{\mathrm{F}} + (\gamma_5\gamma_4\sigma_1 + \alpha_5\gamma_6)\alpha_2\|V^{(t)} - V^{*(t)}\|_{\mathrm{F}}$$

$$+ (\gamma_5\gamma_4\sigma_1 + \alpha_5\gamma_6)\gamma_2\sigma_1\|\overline{U}^{(t)} - \overline{U}^{*(t)}\|_{\mathrm{F}}, \tag{A.19}$$

where (i) comes from (A.17), (ii) comes from (A.18), (iii) comes from (A.16), and (iv) comes from (A.14). Similarly, we can obtain

$$\|V^{(t+1)} - V^{*(t+1)}\|_{\mathrm{F}} \leq \alpha_6\|U^{(t)} - U^{*(t+1)}\|_{\mathrm{F}} + \gamma_6\alpha_2\|V^{(t)} - V^{*(t)}\|_{\mathrm{F}}$$

$$+ \gamma_6\gamma_2\sigma_1\|\overline{U}^{(t)} - \overline{U}^{*(t)}\|_{\mathrm{F}}, \tag{A.20}$$

$$\sigma_1\|\overline{U}^{(t+1)} - \overline{U}^{*(t+1)}\|_{\mathrm{F}} \leq \alpha_4\|U^{(t)} - U^{*(t+1)}\|_{\mathrm{F}} + \gamma_4\alpha_2\|V^{(t)} - V^{*(t)}\|_{\mathrm{F}}$$

$$+ \gamma_4\gamma_2\sigma_1\|\overline{U}^{(t)} - \overline{U}^{*(t)}\|_{\mathrm{F}} \tag{A.21}$$

$$\|U^{(t+0.5)} - U^{*(t+1)}\|_{\mathrm{F}} \leq \alpha_3\|U^{(t)} - U^{*(t+1)}\|_{\mathrm{F}} + \gamma_3\alpha_2\|V^{(t)} - V^{*(t)}\|_{\mathrm{F}}$$

$$+ \gamma_3\gamma_2\sigma_1\|\overline{U}^{(t)} - \overline{U}^{*(t)}\|_{\mathrm{F}}. \tag{A.22}$$

For simplicity, we define

$$\phi_{V^{(t+1)}} = \|V^{(t+1)} - V^{*(t+1)}\|_{\mathrm{F}}, \ \phi_{V^{(t+0.5)}} = \|V^{(t+0.5)} - V^{*(t)}\|_{\mathrm{F}}, \ \phi_{\overline{V}^{(t+1)}} = \sigma_1\|\overline{V}^{(t+1)} - \overline{V}^{*(t+1)}\|_{\mathrm{F}},$$

$$\phi_{U^{(t+1)}} = \|U^{(t+1)} - U^{*(t+2)}\|_{\mathrm{F}}, \ \phi_{U^{(t+0.5)}} = \|U^{(t+0.5)} - U^{*(t+1)}\|_{\mathrm{F}}, \ \phi_{\overline{U}^{(t+1)}} = \sigma_1\|\overline{U}^{(t+1)} - \overline{U}^{*(t+1)}\|_{\mathrm{F}}.$$

Then combining (A.13), (A.14) with (A.19)–(A.22), we obtain

$$\max\left\{\phi_{V^{(t+1)}}, \phi_{V^{(t+0.5)}}, \phi_{\overline{V}^{(t+1)}}, \phi_{U^{(t+1)}}, \phi_{U^{(t+0.5)}}, \phi_{\overline{U}^{(t+1)}}\right\} \leq \beta \max\left\{\phi_{V^{(t)}}, \phi_{U^{(t)}}, \phi_{\overline{U}^{(t)}}\right\}, \tag{A.23}$$

where $\beta$ is a contraction coefficient defined as

$$\beta = \max\{\alpha_5\alpha_6 + \gamma_5\alpha_4, \alpha_6, \alpha_4, \alpha_3\} + \max\{\alpha_1, \alpha_2, (\gamma_5\gamma_4\sigma_1 + \alpha_5\gamma_6), \gamma_6\alpha_2, \gamma_4\alpha_2, \gamma_3\alpha_2\}$$

$$+ \max\{\gamma_1, \gamma_2, (\gamma_5\gamma_4\sigma_1 + \alpha_5\gamma_6)\gamma_2, \gamma_6\gamma_2, \gamma_4\gamma_2, \gamma_3\gamma_2\}.$$

Then we can choose $\xi$ as a sufficiently large constant such that $\beta < 1$. By recursively applying (A.23) for $t = 0, ..., T$, we obtain

$$\max\left\{\phi_{V^{(T)}}, \phi_{V^{(T-0.5)}}, \phi_{\overline{V}^{(T)}}, \phi_{U^{(T)}}, \phi_{U^{(T-0.5)}}, \phi_{\overline{U}^{(T)}}\right\} \leq \beta \max\left\{\phi_{V^{(T-1)}}, \phi_{U^{(T-1)}}, \phi_{\overline{U}^{(T-1)}}\right\}$$

$$\leq \beta^2 \max\left\{\phi_{V^{(T-2)}}, \phi_{U^{(T-2)}}, \phi_{\overline{U}^{(T-2)}}\right\} \leq ... \leq \beta^T \max\left\{\phi_{V^{(0)}}, \phi_{U^{(0)}}, \phi_{\overline{U}^{(0)}}\right\}.$$

By Corollary A.5, we obtain

$$\|U^{(0)} - U^{*(1)}\|_{\mathrm{F}} \leq \frac{3\sigma_1\sqrt{\delta_{2k}}}{\sigma_k}\|V^{(0)} - V^{*(0)}\|_{\mathrm{F}} + \left(\frac{6}{\xi} + 1\right)\sigma_1\|\overline{U}^{(0)} - \overline{U}^{*(0)}\|_{\mathrm{F}}$$

$$\overset{(i)}{\leq} \frac{3\sigma_1}{\sigma_k} \cdot \frac{\sigma_k^3}{12\xi\sigma_1^3} \cdot \frac{\sigma_k^2}{2\xi\sigma_1} + \left(\frac{6}{\xi} + 1\right)\frac{\sigma_k^2}{4\xi\sigma_1}$$

$$\overset{(ii)}{=} \frac{\sigma_k^4}{8\xi^2\sigma_1^3} + \frac{3\sigma_k^2}{2\xi^2\sigma_1} + \frac{\sigma_k^2}{4\xi\sigma_1} \overset{(iii)}{\leq} \frac{\sigma_k^2}{2\xi\sigma_1}, \tag{A.24}$$

where (i) and (ii) come from Lemma A.4, and (iii) comes from the definition of $\xi$ and $\sigma_1 \geq \sigma_k$. Combining (A.24) with Lemma A.4, we have

$$\left\{\phi_{V^{(0)}}, \phi_{U^{(0)}}, \phi_{\overline{U}^{(0)}}\right\} \leq \max\left\{\frac{\sigma_k^2}{2\xi\sigma_1}, \frac{\sigma_k^2}{4\xi\sigma_1^2}\right\}.$$

Then we need at most

$$T = \left\lceil \log\left(\max\left\{\frac{\sigma_k^2}{\xi\sigma_1}, \frac{\sigma_k^2}{2\xi\sigma_1^2}, \frac{\sigma_k^2}{\xi}, \frac{\sigma_k^2}{2\xi\sigma_1}\right\} \cdot \frac{1}{\epsilon}\right) \log^{-1}(\beta^{-1}) \right\rceil$$

iterations such that

$$\|\overline{V}^{(T)} - \overline{V}^*\|_{\mathrm{F}} \leq \beta^T \max\left\{\frac{\sigma_k^2}{2\xi\sigma_1}, \frac{\sigma_k^2}{4\xi\sigma_1^2}\right\} \leq \frac{\epsilon}{2},$$

$$\|U^{(T)} - U^*\|_{\mathrm{F}} \leq \beta^T \max\left\{\frac{\sigma_k^2}{2\xi\sigma_1}, \frac{\sigma_k^2}{4\xi\sigma_1^2}\right\} \leq \frac{\epsilon}{2\sigma_1}.$$

We then follow similar lines to (4.9) in Appendix 4.1, and show $\|M^{(T)} - M^*\|_{\mathrm{F}} \leq \epsilon$, which completes the proof. $\qquad\square$

## A.2 Proof of Theorem 3.4 (Gradient Descent)

*Proof.* The convergence analysis of the gradient descent algorithm is similar to that of the alternating gradient descent. The only difference is that for updating $U$, the gradient descent algorithm employs $V = \overline{V}^{(t)}$ instead of $V = \overline{V}^{(t+1)}$ to calculate the gradient at $U = U^{(t)}$. Then everything else directly follows Appendix A.1, and is therefore omitted.. $\qquad\square$

## B   Lemmas for Theorem 3.4 (Alternating Exact Minimization)

### B.1   Proof of Lemma 4.1

*Proof.* For notational convenience, we omit the index $t$ in $\overline{U}^{*(t)}$ and $V^{*(t)}$, and denote them by $\overline{U}^*$ and $V^*$ respectively. Then we define

$$S^{(t)} = \begin{bmatrix} S_{11}^{(t)} & \cdots & S_{1k}^{(t)} \\ \vdots & \ddots & \vdots \\ S_{k1}^{(t)} & \cdots & S_{kk}^{(t)} \end{bmatrix} \quad \text{with} \quad S_{pq}^{(t)} = \sum_{i=1}^{d} A_i \overline{U}_{*p}^{(t)} \overline{U}_{*q}^{(t)\top} A_i^\top,$$

$$G^{(t)} = \begin{bmatrix} G_{11}^{(t)} & \cdots & G_{1k}^{(t)} \\ \vdots & \ddots & \vdots \\ G_{k1}^{(t)} & \cdots & G_{kk}^{(t)} \end{bmatrix} \quad \text{with} \quad G_{pq}^{(t)} = \sum_{i=1}^{d} A_i \overline{U}_{*p}^* \overline{U}_{*q}^{*\top} A_i^\top$$

for $1 \leq p, q \leq k$. Note that $S^{(t)}$ and $G^{(t)}$ are essentially the partial Hessian matrices $\nabla_V^2 \mathcal{F}(\overline{U}^{(t)}, V)$ and $\nabla_V^2 \mathcal{F}(\overline{U}^*, V)$ for a vectorized $V$, i.e., $\mathrm{vec}(V)$. Before we proceed with the main proof, we first introduce the following lemma.

**Lemma B.1.** Suppose that $\mathcal{A}(\cdot)$ satisfies $2k$-RIP with parameter $\delta_{2k}$. We then have
$$1 + \delta_{2k} \geq \sigma_{\max}(S^{(t)}) \geq \sigma_{\min}(S^{(t)}) \geq 1 - \delta_{2k}.$$

The proof of Lemma B.1 is provided in Appendix B.7. Note that Lemma B.1 is also applicable $G^{(t)}$, since $G^{(t)}$ shares the same structure with $S^{(t)}$. Given a fixed $\overline{U}$, $\mathcal{F}(\overline{U}, V)$ is a quadratic function of $V$. Therefore we have

$$\mathcal{F}(\overline{U}, V') = \mathcal{F}(\overline{U}, V) + \langle \nabla_V \mathcal{F}(\overline{U}, V), V' - V \rangle$$
$$+ \langle \mathrm{vec}(V') - \mathrm{vec}(V), \nabla_V^2 F(\overline{U}, V) (\mathrm{vec}(V') - \mathrm{vec}(V)) \rangle,$$

which further implies implies

$$\mathcal{F}(\overline{U}, V') - \mathcal{F}(\overline{U}, V) - \langle \nabla_V(\overline{U}, V), V' - V \rangle \leq \sigma_{\max}(\nabla_V^2 F(\overline{U}, V)) \|V' - V\|_{\mathrm{F}}^2$$
$$\mathcal{F}(\overline{U}, V') - \mathcal{F}(\overline{U}, V) - \langle \nabla_V(\overline{U}, V), V' - V \rangle \geq \sigma_{\min}(\nabla_V^2 F(\overline{U}, V)) \|V' - V\|_{\mathrm{F}}^2.$$

Then we can verify that $\nabla_V^2 F(U, V)$ also shares the same structure with $S^{(t)}$. Thus applying Lemma B.1 to the above two inequalities, we complete the proof. $\qquad\square$

## B.2   Proof of Lemma 4.3

*Proof.* For notational convenience, we omit the index $t$ in $\overline{U}^{*(t)}$ and $V^{*(t)}$, and denote them by $\overline{U}^*$ and $V^*$ respectively. We define

$$
J^{(t)} = \begin{bmatrix} J_{11}^{(t)} & \cdots & J_{1k}^{(t)} \\ \vdots & \ddots & \vdots \\ J_{k1}^{(t)} & \cdots & J_{kk}^{(t)} \end{bmatrix} \quad \text{with} \quad J_{pq}^{(t)} = \sum_{i=1}^{d} A_i \overline{U}_{*p}^{(t)} \overline{U}_{*q}^{*\top} A_i^\top,
$$

$$
K^{(t)} = \begin{bmatrix} K_{11}^{(t)} & \cdots & K_{1k}^{(t)} \\ \vdots & \ddots & \vdots \\ K_{k1}^{(t)} & \cdots & K_{kk}^{(t)} \end{bmatrix} \quad \text{with} \quad K_{pq}^{(t)} = \overline{U}_{*p}^{(t)\top} \overline{U}_{*q}^* I_n
$$

for $1 \le p, q \le k$. Before we proceed with the main proof, we first introduce the following lemmas.

**Lemma B.2.** Suppose that $\mathcal{A}(\cdot)$ satisfies $2k$-RIP with parameter $\delta_{2k}$. We then have

$$
\|S^{(t)} K^{(t)} - J^{(t)}\|_2 \le \delta_{2k} \sqrt{2k} \|\overline{U}^{(t)} - \overline{U}^*\|_F.
$$

The proof of Lemma B.2 is provided in Appendix B.8. Note that Lemma B.2 is also applicable to $G^{(t)} K^{(t)} - J^{(t)}$, since $G^{(t)}$ and $S^{(t)}$ share the same structure.

**Lemma B.3.** Given $F \in \mathbb{R}^{k \times k}$, we define

$$
\mathbb{F} = \begin{bmatrix} F_{11} I_n & \cdots & F_{1k} I_n \\ \vdots & \ddots & \vdots \\ F_{k1} I_n & \cdots & F_{kk} I_n \end{bmatrix}.
$$

For any $V \in \mathbb{R}^{n \times k}$, let $v = \mathrm{vec}(V)$, then we have $\|\mathbb{F} v\|_2 = \|F V^\top\|_F$.

*Proof.* By linear algebra, we have

$$
[FV]_{ij} = F_{i*}^\top V_{j*} = \sum_{\ell=1}^{k} F_{i\ell} V_{j\ell} = \sum_{\ell=1}^{k} F_{i\ell} I_{*\ell}^\top V_{*\ell},
$$

which completes the proof. $\qquad\square$

We then proceed with the main proof. Since $b_i = \mathrm{tr}(V^{*\top} A_i U^*)$, then we rewrite $\mathcal{F}(\overline{U}, V)$ as

$$
\mathcal{F}(\overline{U}, V) = 1/2 \cdot \sum_{i=1}^{d} \left( \mathrm{tr}(V^\top A_i \overline{U}) - b_i \right)^2 = 1/2 \cdot \sum_{i=1}^{d} \left( \sum_{j=1}^{k} V_{j*}^\top A_i \overline{U}_{*j} - \sum_{j=1}^{k} V_{j*}^{*\top} A_i \overline{U}_{*j}^* \right)^2.
$$

For notational simplicity, we define $v = \mathrm{vec}(V)$. Since $V^{(t+0.5)}$ minimizes $\mathcal{F}(\overline{U}^{(t)}, V)$, we have

$$
\mathrm{vec}\big(\nabla_U \mathcal{F}(\overline{U}^{(t)}, V^{(t+0.5)})\big) = S^{(t)} v^{(t+0.5)} - J^{(t)} v^* = 0.
$$

Solving the above system of equations, we obtain

$$
v^{(t+0.5)} = (S^{(t)})^{-1} J^{(t)} v^*. \tag{B.1}
$$

Meanwhile, we have

$$
\begin{aligned}
\mathrm{vec}(\nabla_V \mathcal{F}(\overline{U}^*, V^{(t+0.5)})) &= G^{(t)} v^{(t+0.5)} - G^{(t)} v^* \\
&= G^{(t)} (S^{(t)})^{-1} J^{(t)} v^* - G^{(t)} v^* = G^{(t)} \big( (S^{(t)})^{-1} J^{(t)} - I_{nk} \big) v^*,
\end{aligned} \tag{B.2}
$$

where the second equality come from (B.1). By triangle inequality, (B.2) further implies

$$
\begin{aligned}
\|((S^{(t)})^{-1} J^{(t)} - I_{nk}) v^*\|_2 &\le \|(K^{(t)} - I_{nk}) v^*\|_2 + \|(S^{(t)})^{-1} (J^{(t)} - S^{(t)} K^{(t)}) v^*\|_2 \\
&\le \|(\overline{U}^{(t)\top} \overline{U}^* - I_k) V^*\|_F + \|(S^{(t)})^{-1}\|_2 \|(J^{(t)} - S^{(t)} K^{(t)}) v^*\|_2 \\
&\le \|\overline{U}^{(t)\top} \overline{U}^* - I_k\|_F \|V^*\|_2 + \|(S^{(t)})^{-1}\|_2 \|(J^{(t)} - S^{(t)} K^{(t)}) v^*\|_2,
\end{aligned} \tag{B.3}
$$

where the second inequality comes from Lemma B.3. Plugging (B.3) into (B.2), we have

$$\|\text{vec}(\nabla_V \mathcal{F}(\overline{U}^*, V^{(t+0.5)}))\|_2 \le \|G^{(t)}\|_2 \|((S^{(t)})^{-1}J^{(t)} - I_{nk})v^*\|_2$$

$$\stackrel{(i)}{\le} (1+\delta_{2k})(\sigma_1 \|\overline{U}^{(t)\top}\overline{U}^* - I_k\|_2 + \|(S^{(t)})^{-1}\|_2 \|S^{(t)}K^{(t)} - J^{(t)}\|_2 \sigma_1 \sqrt{k})$$

$$\stackrel{(ii)}{\le} (1+\delta_{2k})\sigma_1 \left( \|(\overline{U}^{(t)} - \overline{U}^*)^\top (\overline{U}^{(t)} - \overline{U}^*)\|_F + \frac{\sqrt{2}\delta_{2k}k}{1-\delta_{2k}}\|\overline{U}^{(t)} - \overline{U}^*\|_F \right)$$

$$\stackrel{(iii)}{\le} (1+\delta_{2k})\sigma_1 \left( \|\overline{U}^{(t)} - \overline{U}^*\|_F^2 + \frac{\sqrt{2}\delta_{2k}k}{1-\delta_{2k}}\|\overline{U}^{(t)} - \overline{U}^*\|_F \right) \stackrel{(iv)}{\le} \frac{(1-\delta_{2k})\sigma_k}{2\xi}\|\overline{U}^* - \overline{U}^{(t)}\|_F,$$

where (i) comes from Lemma B.1 and $\|V^*\|_2 = \|M^*\| = \sigma_1$ and $\|V^*\|_F = \|v^*\|_2 \le \sigma_1\sqrt{k}$, (ii) comes from Lemmas B.1 and B.2, (iii) from Cauchy-Schwartz inequality, and (iv) comes from (4.1). By Cauchy-Schwartz inequality again, we obtain

$$\mathcal{D}(V^{(t+0.5)}, V^{(t+0.5)}, \overline{U}^{(t)}) \le \|\nabla_V \mathcal{F}(\overline{U}^*, V^{(t+0.5)})\|_F \le \frac{(1-\delta_{2k})\sigma_k}{2\xi}\|\overline{U}^* - \overline{U}^{(t)}\|_F,$$

which completes the proof. $\qquad\square$

## B.3 Proof of Lemma 4.4

*Proof.* For notational convenience, we omit the index $t$ in $\overline{U}^{*(t)}$ and $V^{*(t)}$, and denote them by $\overline{U}^*$ and $V^*$ respectively. By the strong convexity of $\mathcal{F}(\overline{U}^*, \cdot)$, we have

$$\mathcal{F}(\overline{U}^*, V^*) - \frac{1-\delta_{2k}}{2}\|V^{(t+0.5)} - V^*\|_F^2$$

$$\ge \mathcal{F}(\overline{U}^*, V^{(t+0.5)}) + \langle \nabla_V \mathcal{F}(\overline{U}^*, V^{(t+0.5)}), V^* - V^{(t+0.5)} \rangle. \tag{B.4}$$

By the strong convexity of $\mathcal{F}(\overline{U}^*, \cdot)$ again, we have

$$\mathcal{F}(\overline{U}^*, V^{(t+0.5)}) \ge \mathcal{F}(\overline{U}^*, V^*) + \langle \nabla_V \mathcal{F}(\overline{U}^*, V^*), V^{(t+0.5)} - V^{(t+0.5)} \rangle + \frac{1-\delta_{2k}}{2}\|V^{(t+0.5)} - V^*\|_F^2$$

$$\ge \mathcal{F}(\overline{U}^*, V^*) + \frac{1-\delta_{2k}}{2}\|V^{(t+0.5)} - V^*\|_F^2, \tag{B.5}$$

where the last inequality comes from the optimality condition of $V^* = \text{argmin}_V \mathcal{F}(\overline{U}^*, V)$, i.e.

$$\langle \nabla_V \mathcal{F}(\overline{U}^*, V^*), V^{(t+0.5)} - V^* \rangle \ge 0.$$

Meanwhile, since $V^{(t+0.5)}$ minimizes $\mathcal{F}(\overline{U}^{(t)}, \cdot)$, we have the optimality condition

$$\langle \nabla_V \mathcal{F}(\overline{U}^{(t)}, V^{(t+0.5)}), V^* - V^{(t+0.5)} \rangle \ge 0,$$

which further implies

$$\langle \nabla_V \mathcal{F}(\overline{U}^*, V^{(t+0.5)}), V^* - V^{(t+0.5)} \rangle$$

$$\ge \langle \nabla_V \mathcal{F}(\overline{U}^*, V^{(t+0.5)}) - \nabla_V \mathcal{F}(\overline{U}^{(t)}, V^{(t+0.5)}), V^* - V^{(t+0.5)} \rangle. \tag{B.6}$$

Combining (B.4) and (B.5) with (B.6), we obtain

$$\|V^{(t+0.5)} - V^*\|_2 \le \frac{1}{1-\delta_{2k}}\mathcal{D}(V^{(t+0.5)}, V^{(t+0.5)}, \overline{U}^{(t)}),$$

which completes the proof. $\qquad\square$

## B.4 Proof of Lemma 4.5

*Proof.* Before we proceed with the proof, we first introduce the following Lemma

**Lemma B.4.** Suppose that $A^* \in \mathbb{R}^{n \times k}$ is a rank $k$ matrix. Let $E \in \mathbb{R}^{n \times k}$ satisfy $\|E\|_2 \|A^{*\dagger}\|_2 < 1$. Then given a QR decomposition $(A^* + E) = QR$, there exists a factorization of $A^* = Q^*O^*$ such that $Q^* \in \mathbb{R}^{n \times k}$ is an orthonormal matrix, and satisfies

$$\|Q - Q^*\|_F \le \frac{\sqrt{2}\|A^{*\dagger}\|_2 \|E\|_F}{1 - \|E\|_2 \|A^{*\dagger}\|_2}.$$

The proof of Lemma B.4 is provided in [26], therefore omitted.

We then proceed with the main proof. We consider $A^* = V^{*(t)}$ and $E = V^{(t+0.5)} - V^{*(t)}$ in Lemma B.4 respectively. We can verify that

$$\|V^{(t+0.5)} - V^{*(t)}\|_2 \|V^{*(t)\dagger}\|_2 \leq \frac{\|V^{(t+0.5)} - V^{*(t)}\|_{\mathrm{F}}}{\sigma_k} \leq \frac{1}{4}.$$

Then there exists a $V^{*(t)} = \overline{V}^{*(t+1)} O^*$ such that $\overline{V}^{*(t+1)}$ is an orthonormal matrix, and satisfies

$$\|\overline{V}^{*(t+0.5)} - \overline{V}^{*(t+1)}\|_{\mathrm{F}} \leq 2\|V^{*(t)\dagger}\|_2 \|V^{(t+0.5)} - V^{*(t)}\|_{\mathrm{F}} \leq \frac{2}{\sigma_k}\|V^{(t+0.5)} - V^{*(t)}\|_{\mathrm{F}}.$$

Thus we conclude the proof. $\qquad\square$

## B.5 Proof of Lemma 4.6

*Proof.* We first introduce the following lemma.

**Lemma B.5.** Let $b = \mathcal{A}(M^*) + \varepsilon$, $M$ is a rank-$k$ matrix, and $\mathcal{A}$ is a linear measurement operator that satisfies $2k$-RIP with constant $\delta_{2k} < 1/3$. Let $X^{(t+1)}$ be the $(t+1)$-th step iterate of SVP, then we have

$$\|\mathcal{A}(X^{(t+1)}) - b\|_2^2 \leq \|\mathcal{A}(M^*) - b\|_2^2 + 2\delta_{2k}\|\mathcal{A}(X^{(t)}) - b\|_2^2$$

The proof of Lemma B.5 is provided in [12], therefore omitted. We then explain the implication of Lemma B.5. [12] show that $X^{(t+1)}$ is obtained by taking a projected gradient iteration over $X^{(t)}$ using step size $\frac{1}{1+\delta_{2k}}$. Then taking $X^{(t)} = 0$, we have

$$X^{(t+1)} = \frac{\overline{U}^{(0)}\overline{\Sigma}^{(0)}\overline{V}^{(0)\top}}{1 + \delta_{2k}}.$$

Suppose that $M^*$ has a compact singular value decomposition $M^* = \widetilde{U}^* \widetilde{D}^* \widetilde{V}^{*\top}$. Then Lemma B.5 implies

$$\left\|\mathcal{A}\left(\frac{\overline{U}^{(0)}\overline{\Sigma}^{(0)}\overline{V}^{(0)\top}}{1 + \delta_{2k}} - \widetilde{U}^* \widetilde{D}^* \widetilde{V}^{*\top}\right)\right\|_2^2 \leq 4\delta_{2k}\|\mathcal{A}(\widetilde{U}^* \widetilde{D}^* \widetilde{V}^{*\top})\|_2^2. \tag{B.7}$$

Since $\mathcal{A}(\cdot)$ satisfies $2k$-RIP, (B.7) further implies

$$\left\|\frac{\overline{U}^{(0)}\overline{\Sigma}^{(0)}\overline{V}^{(0)\top}}{1 + \delta_{2k}} - \widetilde{U}^* \widetilde{D}^* \widetilde{V}^{*\top}\right\|_{\mathrm{F}}^2 \leq 4\delta_{2k}(1 + 3\delta_{2k})\|\widetilde{D}^*\|_{\mathrm{F}}^2. \tag{B.8}$$

We then project each column of $\widetilde{U}^* \widetilde{D}^* \widetilde{V}^{*\top}$ into the subspace spanned by $\{\overline{U}_{*i}^{(0)}\}_{i=1}^k$, and obtain

$$\|\overline{U}^{(0)}\overline{U}^{(0)\top}\widetilde{U}^* \widetilde{D}^* \widetilde{V}^{*\top} - \widetilde{U}^* \widetilde{D}^* \widetilde{V}^{*\top}\|_{\mathrm{F}}^2 \leq 6\delta_{2k}\|\widetilde{D}^*\|_{\mathrm{F}}^2.$$

Let $\overline{U}_\perp^{(0)}$ denote the orthonormal complement of $\overline{U}^{(0)}$, i.e.,

$$\overline{U}_\perp^{(0)\top}\overline{U}_\perp^{(0)} = I_{n-k} \quad \text{and} \quad \overline{U}_\perp^{(0)\top}\overline{U}^{(0)} = 0.$$

Then we have

$$\frac{6\delta_{2k}k\sigma_1^2}{\sigma_k^2} \geq \|(\overline{U}^{(0)}\overline{U}^{(0)\top} - I_n)\widetilde{U}^*\|_{\mathrm{F}}^2 = \|\overline{U}_\perp^{(0)\top}\widetilde{U}^*\|_{\mathrm{F}}^2.$$

Thus there exists a unitary matrix $O \in \mathbb{R}^{k \times k}$ such that $OO^\top = I_k$ and

$$\|\overline{U}^{(0)} - \widetilde{U}^*O\|_{\mathrm{F}} \leq \sqrt{2}\|\overline{U}_\perp^{(0)\top}\widetilde{U}^*\|_{\mathrm{F}} \leq 2\sqrt{3\delta_{2k}k} \cdot \frac{\sigma_1}{\sigma_k}.$$

We define $\overline{U}^{*(0)} = \widetilde{U}^*O$. Then combining the above inequality with (4.3), we have

$$\|\overline{U}^{(0)} - \overline{U}^{*(0)}\|_{\mathrm{F}} \leq \frac{(1 - \delta_{2k})\sigma_k}{4\xi(1 + \delta_{2k})\sigma_1}.$$

Moreover, we define $V^{*(0)} = \widetilde{V}^* \widetilde{D}^* O$. Then we have $\overline{U}^{*(0)}V^{*(0)\top} = \widetilde{U}^*OO^\top \widetilde{D}^* \widetilde{V}^* = M^*$. $\qquad\square$

## B.6 Proof of Corollary 4.7

*Proof.* Since (4.4) ensures that (4.1) of Lemma 4.3 holds, then we have

$$\|V^{(t+0.5)} - V^{*(t)}\|_{\mathrm{F}} \leq \frac{1}{1-\delta_{2k}} \mathcal{D}(V^{(t+0.5)}, V^{(t+0.5)}, \overline{U}^{(t)})$$

$$\overset{(i)}{\leq} \frac{1}{1-\delta_{2k}} \cdot \frac{(1-\delta_{2k})\sigma_k}{2\xi} \|\overline{U}^{(t)} - \overline{U}^{*(t)}\|_{\mathrm{F}}$$

$$\overset{(ii)}{\leq} \frac{1}{1-\delta_{2k}} \cdot \frac{(1-\delta_{2k})\sigma_k}{2\xi} \cdot \frac{(1-\delta_{2k})\sigma_k}{4\xi(1+\delta_{2k})\sigma_1}$$

$$\leq \left( \frac{(1-\delta_{2k})\sigma_k}{8\xi^2(1+\delta_{2k})\sigma_1} \right) \sigma_k \overset{(iii)}{\leq} \frac{\sigma_k}{4}, \tag{B.9}$$

where (i) comes from Lemma 4.4, (ii) comes from (4.4), and (iii) comes from the definition of $\xi$ and $\sigma_k \leq \sigma_1$. Since (B.9) ensures that (4.2) of Lemma 4.5 holds for $V^{(t+0.5)}$, we obtain

$$\|\overline{V}^{(t+1)} - \overline{V}^{*(t+1)}\|_{\mathrm{F}} \leq \frac{2}{\sigma_k} \|V^{(t+0.5)} - V^{*(t)}\|_{\mathrm{F}} \overset{(i)}{\leq} \frac{1}{\xi} \|\overline{U}^{(t)} - \overline{U}^{*(t)}\|_{\mathrm{F}} \overset{(ii)}{\leq} \frac{(1-\delta_{2k})\sigma_k}{4\xi(1+\delta_{2k})\sigma_1}, \tag{B.10}$$

where (i) comes from (B.9), and (ii) comes from the definition of $\xi$ and (4.4). $\qquad\square$

## B.7 Proof of Lemma B.1

*Proof.* We consider an arbitrary $W \in \mathbb{R}^{n \times k}$ such that $\|W\|_{\mathrm{F}} = 1$. Let $w = \mathrm{vec}(W)$. Then have

$$w^\top B w = \sum_{p,q=1}^k W_{*p}^\top S_{pq}^{(t)} W_{*p} = \sum_{p,q=1}^k W_{*p}^\top \left( \sum_{i=1}^d A_i \overline{U}_{*p}^{(t)} \overline{U}_{*q}^{(t)\top} A_i^\top \right) W_{*q}$$

$$= \sum_{i=1}^d \left( \sum_{p=1}^k W_{*p}^\top A_i \overline{U}_{*p}^{(t)} \right) \left( \sum_{q=1}^k W_{*q}^\top A_i \overline{U}_{*q}^{(t)} \right) = \sum_{i=1}^n \mathrm{tr}(W^\top A_i \overline{U}^{(t)})^2.$$

Since $\mathcal{A}(\cdot)$ satisfies $2k$-RIP, then we have

$$\sum_{i=1}^d \mathrm{tr}(W^\top A_i \overline{U}^{(t)})^2 \geq (1-\delta_{2k}) \|\overline{U}^{(t)} W^\top\|_{\mathrm{F}} = (1-\delta_{2k}) \|W\|_{\mathrm{F}} = 1 - \delta_{2k},$$

$$\sum_{i=1}^d \mathrm{tr}(W^\top A_i \overline{U}^{(t)})^2 \leq (1+\delta_{2k}) \|\overline{U}^{(t)} W^\top\|_{\mathrm{F}} = (1+\delta_{2k}) \|W\|_{\mathrm{F}} = 1 + \delta_{2k}.$$

Since $W$ is arbitrary, then we have

$$\sigma_{\min}(S^{(t)}) = \min_{\|w\|_2=1} w^\top S^{(t)} w \geq 1 - \delta_{2k} \quad \text{and} \quad \sigma_{\max}(S^{(t)}) = \max_{\|w\|_2=1} w^\top S^{(t)} w \leq 1 + \delta_{2k}.$$

Thus we conclude the proof. $\qquad\square$

## B.8 Proof of Lemma B.2

*Proof.* For notational convenience, we omit the index $t$ in $\overline{U}^{*(t)}$ and $V^{*(t)}$, and denote them by $\overline{U}^*$ and $V^*$ respectively. We first introduce the following lemma.

**Lemma B.6.** Suppose $\mathcal{A}(\cdot)$ satisfies $2k$-RIP. For any $U$, $U' \in \mathbb{R}^{m \times k}$ and $V$, $V' \in \mathbb{R}^{n \times k}$, we have

$$|\langle \mathcal{A}(UV^\top), \mathcal{A}(U'V'^\top) \rangle - \langle U^\top U', V^\top V' \rangle| \leq 3\delta_{2k} \|UV^\top\|_{\mathrm{F}} \cdot \|U'V'^\top\|_{\mathrm{F}}.$$

The proof of Lemma B.6 is provided in [14], and hence omitted.

We now proceed with the proof. We consider arbitrary $W, Z \in \mathbb{R}^{n \times k}$ such that $\|W\|_{\mathrm{F}} = \|Z\|_{\mathrm{F}} = 1$. Let $w = \mathrm{vec}(W)$ and $z = \mathrm{vec}(Z)$. Then have

$$w^\top (S^{(t)} K^{(t)} - J^{(t)}) z = \sum_{p,q=1}^k W_{*p}^\top [S^{(t)} K^{(t)} - J^{(t)}]_{pq} Z_{*q}.$$

We consider a decomposition

$$[S^{(t)}K^{(t)} - J^{(t)}]_{pq} = \sum_{\ell=1}^{k} S_{p\ell}^{(t)} K_{\ell q}^{(t)} - J_{pq}^{(t)} = \sum_{\ell=1}^{k} S_{p\ell}^{(t)} \overline{U}_{*\ell}^{(t)\top} \overline{U}_{*q}^{*} I_n - J_{pq}^{(t)}$$

$$= \sum_{\ell=1}^{k} \overline{U}_{*q}^{*\top} \overline{U}_{*\ell}^{(t)} \sum_{i=1}^{d} A_i \overline{U}_{*p}^{(t)} \overline{U}_{*\ell}^{(t)} A_i^{\top} - J_{pq}^{(t)} = \sum_{\ell=1}^{k} A_i \overline{U}_{*q}^{*\top} \overline{U}_{*\ell}^{(t)} \sum_{i=1}^{d} \overline{U}_{*p}^{(t)} \overline{U}_{*\ell}^{(t)} A_i^{\top} - \sum_{i=1}^{d} A_i \overline{U}_{*p}^{(t)} \overline{U}_{*q}^{*} A_i^{\top}$$

$$= \sum_{i=1}^{d} A_i \overline{U}_{*p}^{(t)} \overline{U}_{*q}^{*} (\overline{U}^{(t)} \overline{U}^{(t)\top} - I_n) A_i^{\top}.$$

which further implies

$$w^{\top}(S^{(t)}K^{(t)} - J^{(t)})z = \sum_{p,q} W_{*p}^{\top} \left( \sum_{i=1}^{d} A_i \overline{U}_{*p}^{(t)} \overline{U}_{*q}^{*} (\overline{U}^{(t)} \overline{U}^{(t)\top} - I_n) A_i^{\top} \right) Z_{*q}$$

$$= \sum_{i=1}^{d} \sum_{p,q} W_{*p}^{\top} A_i \overline{U}_{*p}^{(t)} \overline{U}_{*q}^{*} (\overline{U}^{(t)} \overline{U}^{(t)\top} - I_n) A_i^{\top} Z_{*q}$$

$$= \sum_{i=1}^{d} \operatorname{tr}(W^{\top} A_i \overline{U}^{(t)}) \operatorname{tr}\left( Z^{\top} A_i (\overline{U}^{(t)} \overline{U}^{(t)\top} - I_n) \overline{U}^{*} \right).$$

Therefore by $2k$-RIP of $\mathcal{A}(\cdot)$ and Lemma B.6, we obtain

$$w^{\top}(S^{(t)}K^{(t)} - J^{(t)})z$$

$$\leq \operatorname{tr}\left( \overline{U}^{*} (\overline{U}^{(t)} \overline{U}^{(t)\top} - I_n) \overline{U}^{(t)} W^{\top} Z \right) + \delta_{2k} \|\overline{U}^{(t)} W^{\top}\|_{\mathrm{F}} \|(\overline{U}^{(t)} \overline{U}^{(t)\top} - I_n) \overline{U}^{*} Z^{\top}\|_{\mathrm{F}}$$

$$\leq \delta_{2k} \|W\|_{\mathrm{F}} \sqrt{\|\overline{U}^{*\top} (\overline{U}^{(t)} \overline{U}^{(t)\top} - I_n) \overline{U}^{*}\|_{\mathrm{F}} \|Z^{\top} Z\|_{\mathrm{F}}} \leq \delta_{2k} \sqrt{2k} \|\overline{U}^{(t)} - \overline{U}^{*}\|_{\mathrm{F}},$$

where the last inequality comes from $(\overline{U}^{(t)} \overline{U}^{(t)\top} - I_n) \overline{U}^{(t)} = 0$. Since $W$ and $Z$ are arbitrary, we have

$$\sigma_{\max}(S^{(t)}K^{(t)} - J^{(t)}) = \max_{\|w\|_2=1, \|z\|_2=1} w^{\top}(S^{(t)}K^{(t)} - J^{(t)})w \leq \delta_{2k} \sqrt{2k} \|\overline{U}^{(t)} - \overline{U}^{*}\|_{\mathrm{F}},$$

which completes the proof. $\qquad\square$

# C   Lemmas for Theorem 3.4 (Alternating Gradient Descent)

## C.1   Proof of Lemma A.1

*Proof.* For notational convenience, we omit the index $t$ in $\overline{U}^{*(t)}$ and $V^{*(t)}$, and denote them by $\overline{U}^{*}$ and $V^{*}$ respectively. We have

$$\operatorname{vec}(\nabla_V \mathcal{F}(\overline{U}^{(t)}, V^{(t)})) = S^{(t)} v^{(t)} - J^{(t)} v^{*} \quad \text{and} \quad \operatorname{vec}(\nabla_V \mathcal{F}(\overline{U}^{*}, V^{(t)})) = G^{(t)} v^{(t)} - G^{(t)} v^{*}.$$

Therefore, we further obtain

$$\|\nabla_V \mathcal{F}(\overline{U}^{(t)}, V^{(t)}) - \nabla_V \mathcal{F}(\overline{U}^{*}, V^{(t)})\|_{\mathrm{F}}$$

$$= \|(S^{(t)} - J^{(t)})(v^{(t)} - v^{*}) + (S^{(t)} - J^{(t)})v^{*} + (J^{(t)} - G^{(t)})(v^{(t)} - v^{*})\|_2$$

$$\leq \|(S^{(t)} - J^{(t)})(v^{(t)} - v^{*})\|_2 + \|(S^{(t)} - J^{(t)})v^{*}\|_2 + \|(J^{(t)} - G^{(t)})(v^{(t)} - v^{*})\|_2$$

$$\leq \|S^{(t)}\|_2 \|((S^{(t)})^{-1} J^{(t)} - I_{nk})(v^{(t)} - v^{*})\|_2$$

$$+ \|S^{(t)}\|_2 \|((S^{(t)})^{-1} J^{(t)} - I_{nk})v^{*}\|_2 + \|G\|_2 \|((G^{(t)})^{-1} J^{(t)} - I_{nk})(v^{(t)} - v^{*})\|_2. \quad \text{(C.1)}$$

Recall that Lemma B.2 is also applicable to $G^{(t)} K^{(t)} - J^{(t)}$. Since we have

$$\|V^{(t)} - V^{*}\|_2 \leq \|V^{(t)} - V^{*}\|_{\mathrm{F}} = \|v^{(t)} - v^{*}\|_2 \leq \sigma_1,$$

following similar lines to Appendix B.2, we can show

$$\|((S^{(t)})^{-1}J^{(t)} - I_{mn})v^*\|_2 \leq \sigma_1\left(\|\overline{U}^{(t)} - \overline{U}^*\|_F^2 + \frac{\sqrt{2}\delta_{2k}k}{1 - \delta_{2k}}\|\overline{U}^{(t)} - \overline{U}^*\|_F\right),$$

$$\|((G^{(t)})^{-1}J^{(t)} - I_{mn})(v^{(t)} - v^*)\|_2 \leq \sigma_1\left(\|\overline{U}^{(t)} - \overline{U}^*\|_F^2 + \frac{\sqrt{2}\delta_{2k}k}{1 - \delta_{2k}}\|\overline{U}^{(t)} - \overline{U}^*\|_F\right),$$

$$\|((S^{(t)})^{-1}J^{(t)} - I_{mn})(v^{(t)} - v^*)\|_2 \leq \sigma_1\left(\|\overline{U}^{(t)} - \overline{U}^*\|_F^2 + \frac{\sqrt{2}\delta_{2k}k}{1 - \delta_{2k}}\|\overline{U}^{(t)} - \overline{U}^*\|_F\right).$$

Combining the above three inequalities with (C.1), we have

$$\|\nabla_V \mathcal{F}(\overline{U}^{(t)}, V^{(t)}) - \nabla_V \mathcal{F}(\overline{U}^*, V^{(t)})\|_F$$
$$\leq 2(1 + \delta_{2k})\sigma_1\left(\|\overline{U}^{(t)} - \overline{U}^*\|_F^2 + \frac{\sqrt{2}\delta_{2k}k}{1 - \delta_{2k}}\|\overline{U}^{(t)} - \overline{U}^*\|_F\right). \tag{C.2}$$

Since $\overline{U}^{(t)}$, $\delta_{2k}$, and $\xi$ satisfy (A.1), then (C.2) further implies

$$\|\nabla_V \mathcal{F}(\overline{U}^{(t)}, V^{(t)}) - \nabla_V \mathcal{F}(\overline{U}^*, V^{(t)})\|_F \leq \frac{(1 + \delta_{2k})\sigma_k}{\xi}\|\overline{U}^{(t)} - \overline{U}^*\|_F. \tag{C.3}$$

Therefore by Cauchy-Schwartz inequality, (C.3) implies

$$\mathcal{D}(V^{(t+0.5)}, V^{(t)}, \overline{U}^{(t)}) \leq \|\nabla_V \mathcal{F}(\overline{U}^{(t)}, V^{(t)}) - \nabla_V \mathcal{F}(\overline{U}^*, V^{(t)})\|_F \leq \frac{(1 + \delta_{2k})\sigma_k}{\xi}\|\overline{U}^{(t)} - \overline{U}^*\|_F,$$

which completes the proof. $\qquad\square$

## C.2 Proof of Lemma A.2

*Proof.* For notational convenience, we omit the index $t$ in $\overline{U}^{*(t)}$ and $V^{*(t)}$, and denote them by $\overline{U}^*$ and $V^*$ respectively. By the strong convexity of $\mathcal{F}(\overline{U}^*, \cdot)$, we have

$$\mathcal{F}(\overline{U}^*, V^*) - \frac{1 - \delta_{2k}}{2}\|V^{(t)} - V^*\|_F^2 \geq \mathcal{F}(\overline{U}^*, V^{(t)}) + \langle\nabla_V \mathcal{F}(\overline{U}^*, V^{(t)}), V^* - V^{(t)}\rangle$$
$$= \mathcal{F}(\overline{U}^*, V^{(t)}) + \langle\nabla_V \mathcal{F}(\overline{U}^*, V^{(t)}), V^{(t+0.5)} - V^{(t)}\rangle + \langle\nabla_V \mathcal{F}(\overline{U}^*, V^{(t)}), V^* - V^{(t+0.5)}\rangle. \tag{C.4}$$

Meanwhile, we define

$$\mathcal{Q}(V; \overline{U}^*, V^{(t)}) = \mathcal{F}(\overline{U}^*, V^{(t)}) + \langle\nabla_V \mathcal{F}(\overline{U}^*, V^{(t)}), V - V^{(t)}\rangle + \frac{1}{2\eta}\|V - V^{(t)}\|_F^2.$$

Since $\eta$ satisfies (A.2) and $\mathcal{F}(\overline{U}^*, V)$ is strongly smooth in $V$ for a fixed orthonormal $\overline{U}^*$, we have
$$\mathcal{Q}(V; \overline{U}^*, V^{(t)}) \geq \mathcal{F}(\overline{U}^*, V^{(t)}).$$

Combining the above two inequalities, we obtain
$$\mathcal{F}(\overline{U}^*, V^{(t)}) + \langle\nabla_V \mathcal{F}(\overline{U}^*, V^{(t)}), V^{(t+0.5)} - V^{(t)}\rangle$$
$$= \mathcal{Q}(V^{(t+0.5)}; \overline{U}^*, V^{(t)}) - \frac{1}{2\eta}\|V^{(t+0.5)} - V^{(t)}\|_F^2 \geq \mathcal{F}(\overline{U}^*, V^{(t+0.5)}) - \frac{1}{2\eta}\|V^{(t+0.5)} - V^{(t)}\|_F^2. \tag{C.5}$$

Moreover, by the strong convexity of $\mathcal{F}(\overline{U}^*, \cdot)$ again, we have

$$\mathcal{F}(\overline{U}^*, V^{(t+0.5)}) \geq \mathcal{F}(\overline{U}^*, V^*) + \langle\nabla_V \mathcal{F}(\overline{U}^*, V^*), V^{(t+0.5)} - V^*\rangle + \frac{1 - \delta_{2k}}{2}\|V^{(t+0.5)} - V^*\|_F^2$$
$$\geq \mathcal{F}(\overline{U}^*, V^*) + \frac{1 - \delta_{2k}}{2}\|V^{(t+0.5)} - V^*\|_F^2, \tag{C.6}$$

where the second equalities comes from the optimality condition of $V^* = \mathrm{argmin}_V \mathcal{F}(\overline{U}^*, V)$, i.e.
$$\langle\nabla_V \mathcal{F}(\overline{U}^*, V^*), V^{(t+0.5)} - V^*\rangle \geq 0.$$

Combining (C.4) and (C.5) with (C.6), we obtain
$$\mathcal{F}(\overline{U}^*, V^{(t)}) + \langle\nabla_V \mathcal{F}(\overline{U}^*, V^{(t)}), V^{(t+0.5)} - V^{(t)}\rangle$$
$$\geq \mathcal{F}(\overline{U}^*, V^*) + \frac{1 - \delta_{2k}}{2}\|V^{(t+0.5)} - V^*\|_F^2 - \frac{1}{2\eta}\|V^{(t+0.5)} - V^{(t)}\|_F^2. \tag{C.7}$$

On the other hand, since $V^{(t+0.5)}$ minimizes $\mathcal{Q}(V; \overline{U}^*, V^{(t)})$, we have

$$0 \leq \langle \nabla \mathcal{Q}(V^{(t+0.5)}; \overline{U}^*, V^{(t)}), V^* - V^{(t+0.5)} \rangle$$

$$\leq \langle \nabla_V \mathcal{F}(\overline{U}^*, V^{(t)}), V^* - V^{(t+0.5)} \rangle + (1 + \delta_{2k}) \langle V^{(t+0.5)} - V^{(t)}, V^* - V^{(t+0.5)} \rangle. \quad \text{(C.8)}$$

Meanwhile, we have

$$\langle \nabla_V \mathcal{F}(\overline{U}^*, V^{(t)}), V^* - V^{(t+0.5)} \rangle$$

$$= \langle \nabla_V \mathcal{F}(\overline{U}^{(t)}, V^{(t)}), V^* - V^{(t+0.5)} \rangle - \mathcal{D}(V^{(t+0.5)}, V^{(t)}, \overline{U}^{(t)}) \|V^* - V^{(t+0.5)}\|_2$$

$$\geq (1 + \delta_{2k}) \langle V^{(t)} - V^{(t+0.5)}, V^* - V^{(t+0.5)} \rangle - \mathcal{D}(V^{(t+0.5)}, V^{(t)}, \overline{U}^{(t)}) \|V^* - V^{(t+0.5)}\|_2$$

$$= (1 + \delta_{2k}) \langle V^{(t)} - V^{(t+0.5)}, V^* - V^{(t)} \rangle + \frac{1}{2\eta} \|V^{(t)} - V^{(t+0.5)}\|_F^2$$

$$- \mathcal{D}(V^{(t+0.5)}, V^{(t)}, \overline{U}^{(t)}) \|V^* - V^{(t+0.5)}\|_2. \quad \text{(C.9)}$$

Combining (C.8) with (C.9), we obtain

$$2 \langle V^{(t)} - V^{(t+0.5)}, V^* - V^{(t)} \rangle$$

$$\leq -\eta(1 - \delta_{2k}) \|V^{(t)} - V^*\|_2^2 - \eta(1 - \delta_{2k}) \|V^{(t+0.5)} - V^*\|_2^2$$

$$- \|V^{(t+0.5)} - V^{(t)}\|_2^2 + \mathcal{D}(V^{(t+0.5)}, V^{(t)}, \overline{U}^{(t)}) \|V^* - V^{(t+0.5)}\|_2. \quad \text{(C.10)}$$

Therefore, combining (C.7) with (C.10), we obtain

$$\|V^{(t+0.5)} - V^*\|_F^2 \leq \|V^{(t+0.5)} - V^{(t)} + V^{(t)} - V^*\|_F^2$$

$$= \|V^{(t+0.5)} - V^{(t)}\|_F^2 + \|V^{(t)} - V^*\|_F^2 + 2 \langle V^{(t+0.5)} - V^{(t)}, V^{(t)} - V^* \rangle$$

$$\leq 2\eta \|V^{(t)} - V^*\|_F^2 - \eta(1 - \delta_{2k}) \|V^{(t+0.5)} - V^*\|_F^2$$

$$- \mathcal{D}(V^{(t+0.5)}, V^{(t)}, \overline{U}^{(t)}) \|V^* - V^{(t+0.5)}\|_2.$$

Rearranging the above inequality, we obtain

$$\|V^{(t+0.5)} - V^*\|_F \leq \sqrt{\delta_{2k}} \|V^{(t)} - V^*\|_F + \frac{2}{1 + \delta_{2k}} \mathcal{D}(V^{(t+0.5)}, V^{(t)}, \overline{U}^{(t)}),$$

which completes the proof. □

## C.3 Proof of Lemma A.3

*Proof.* Before we proceed with the proof, we first introduce the following lemma.

**Lemma C.1.** For any matrix $U, \widetilde{U} \in \mathbb{R}^{m \times k}$ and $V, \widetilde{V} \in \mathbb{R}^{n \times k}$, we have

$$\|UV^\top - \widetilde{U}\widetilde{V}^\top\|_F \leq \|U\|_2 \|V - \widetilde{V}\| + \|\widetilde{V}\|_2 \|U - \widetilde{U}\|_F.$$

*Proof.* By linear algebra, we have

$$\|UV^\top - \widetilde{U}\widetilde{V}^\top\|_F = \|UV^\top - U\widetilde{V}^\top + U\widetilde{V}^\top - \widetilde{U}\widetilde{V}^\top\|_F$$

$$\leq \|UV^\top - U\widetilde{V}^\top\|_F + \|U\widetilde{V}^\top - \widetilde{U}\widetilde{V}^\top\|_F \leq \|U\|_2 \|V - \widetilde{V}\|_F + \|\widetilde{V}\|_2 \|U - \widetilde{U}\|_F. \quad \text{(C.11)}$$

Thus, we conclude the proof. □

By Lemma C.1, we have

$$\|R_{\overline{V}}^{(t+0.5)} - \overline{V}^{*(t+1)\top} V^{*(t)}\|_F = \|\overline{V}^{(t+0.5)\top} V^{(t+0.5)} - \overline{V}^{*(t+1)\top} V^{*(t)}\|_F$$

$$\leq \|\overline{V}^{(t+0.5)}\|_2 \|V^{(t+0.5)} - V^{*(t)}\|_F + \|V^{*(t)}\|_2 \|\overline{V}^{(t+0.5)} - \overline{V}^{*(t+1)}\|_F$$

$$\leq \|V^{(t+0.5)} - V^{*(t)}\|_F + \frac{2\sigma_1}{\sigma_k} \|V^{(t+0.5)} - V^{*(t)}\|_F, \quad \text{(C.12)}$$

where the last inequality comes from Lemma 4.5. Moreover, let $U^{*(t+1)} = \overline{U}^{*(t)} (\overline{V}^{*(t+1)\top} V^{*(t)})^\top$. Then we can verify

$$U^{*(t+1)} \overline{V}^{*(t+1)} = \overline{U}^{*(t)} V^{*(t)\top} \overline{V}^{*(t+1)} \overline{V}^{*(t+1)\top} = M^* \overline{V}^{*(t+1)} \overline{V}^{*(t+1)\top} = M^*,$$

where the last equality holds, since $\overline{V}^{*(t+1)}\overline{V}^{*(t+1)\top}$ is exactly the projection matrix for the row space of $M^*$. Thus by Lemma C.1, we have

$$\|U^{(t+1)} - U^{*(t+1)}\|_{\mathrm{F}} = \|\overline{U}^{(t)}R_{\overline{V}}^{(t+0.5)\top} - \overline{U}^{*(t)}(\overline{V}^{*(t+1)\top}V^{*(t)})^\top\|_{\mathrm{F}}$$

$$\leq \|\overline{U}^{(t)}\|_2\|R_{\overline{V}}^{(t+0.5)} - \overline{V}^{*(t+1)\top}V^{*(t)}\|_{\mathrm{F}} + \|\overline{V}^{*(t+1)\top}V^{*(t)}\|_2\|\overline{U}^{(t)} - \overline{U}^{*(t)}\|_{\mathrm{F}}$$

$$\leq \left(1 + \frac{2\sigma_1}{\sigma_k}\right)\|V^{(t+0.5)} - V^{*(t)}\|_{\mathrm{F}} + \sigma_1\|\overline{U}^{(t)} - \overline{U}^{*(t)}\|_{\mathrm{F}},$$

where the last inequality comes from (C.12), $\|\overline{V}^{*(t+1)}\|_2 = \sigma_1$, and $\|\overline{U}^{(t)}\|_2 = 1$. □

## C.4  Proof of Lemma A.4

*Proof.* Following similar lines to Appendix B.5, we have

$$\|\overline{U}^{(0)} - \overline{U}^{*(0)}\|_{\mathrm{F}} \leq \frac{\sigma_k^2}{4\xi\sigma_1^2}.$$

In Appendix B.5, we have already shown

$$\left\|\frac{\overline{U}^{(0)}\overline{\Sigma}^{(0)}\overline{V}^{(0)\top}}{1 + \delta_{2k}} - M^*\right\|_{\mathrm{F}} \leq 2\sqrt{\delta_{2k}(1 + 3\delta_{2k})}\|\overline{\Sigma}^*\|_{\mathrm{F}}.$$

Then by Lemma C.1 we have

$$\left\|\frac{\overline{U}^{(0)}\overline{\Sigma}^{(0)}}{1 + \delta_{2k}} - V^{*(0)}\right\|_{\mathrm{F}} = \left\|\frac{\overline{U}^{(0)\top}\overline{U}^{(0)}\overline{\Sigma}^{(0)}\overline{V}^{(0)\top}}{1 + \delta_{2k}} - \overline{U}^{*(0)\top}M^*\right\|_{\mathrm{F}}$$

$$\leq \|\overline{U}^{(0)}\|_2\left\|\frac{\overline{U}^{(0)}\overline{\Sigma}^{(0)}\overline{V}^{(0)\top}}{1 + \delta_{2k}} - M^*\right\|_{\mathrm{F}} + \|M^*\|_2\|\overline{U}^{(0)} - \overline{U}^{*(0)}\|_{\mathrm{F}}$$

$$\leq 2\sqrt{\delta_{2k}k(1 + 3\delta_{2k})}\sigma_1 + \frac{\sigma_k^2}{4\xi\sigma_1^1}. \tag{C.13}$$

By triangle inequality, we further have

$$\|\overline{U}^{(0)}\overline{\Sigma}^{(0)} - V^{*(0)}\|_{\mathrm{F}} \leq (1 + \delta_{2k})\left\|\frac{\overline{U}^{(0)}\overline{\Sigma}^{(0)}}{1 + \delta_{2k}} - V^{*(0)}\right\|_{\mathrm{F}} + \delta_{2k}\|V^{*(0)}\|_{\mathrm{F}}$$

$$\overset{(i)}{\leq}(1 + \delta_{2k})\left(2\sqrt{\delta_{2k}k(1 + 3\delta_{2k})}\sigma_1 + \frac{\sigma_k^2}{4\xi\sigma_1}\right) + \delta_{2k}\sigma_1\sqrt{k}$$

$$\overset{(ii)}{\leq}\left(\frac{\sigma_k^3}{9\sigma_1^3\xi} + \frac{\sigma_k^2}{3\sigma_1^3\xi^2} + \frac{\sigma_k^3}{192\xi^3\sigma_1^2}\right)\sigma_1 \overset{(iii)}{\leq} \frac{\sigma_k^2}{2\xi\sigma_1},$$

where (i) comes from (C.13) and $\|V^{*(0)}\|_{\mathrm{F}} = \|M^*\|_{\mathrm{F}} \leq \sigma_1\sqrt{k}$, (ii) comes from (A.4), and (iii) comes from the definition of $\xi$ and $\sigma_1 \geq \sigma_k$. □

## C.5  Proof of Corollary A.5

*Proof.* Since (A.5) ensures that (A.1) of Lemma A.1 holds, we have

$$\|V^{(t+0.5)} - V^{*(t)}\|_{\mathrm{F}} \leq \sqrt{\delta_{2k}}\|V^{(t)} - V^{*(t)}\|_{\mathrm{F}} + \frac{2}{1 + \delta_{2k}}\mathcal{D}(V^{(t+0.5)}, V^{(t)}, \overline{U}^{(t)})$$

$$\overset{(i)}{\leq} \sqrt{\delta_{2k}}\|V^{(t)} - V^{*(t)}\|_{\mathrm{F}} + \frac{2}{1 + \delta_{2k}} \cdot \frac{(1 + \delta_{2k})\sigma_k}{\xi}\|\overline{U}^{(t)} - \overline{U}^{*(t)}\|_{\mathrm{F}}$$

$$\overset{(ii)}{\leq} \frac{\sigma_k^2}{12\xi\sigma_1^2}\|V^{(t)} - V^{*(t)}\|_{\mathrm{F}} + \frac{2\sigma_k}{\xi}\|\overline{U}^{(t)} - \overline{U}^{*(t)}\|_{\mathrm{F}}$$

$$\overset{(iii)}{\leq} \frac{\sigma_k^2}{12\xi\sigma_1^2} \cdot \frac{\sigma_k^2}{2\xi\sigma_1} + \frac{2\sigma_k}{\xi} \cdot \frac{\sigma_k^2}{4\xi\sigma_1^2} \overset{(iv)}{\leq} \frac{13\sigma_k^3}{24\xi^2\sigma_1^2} \overset{(v)}{\leq} \frac{\sigma_k}{4}, \tag{C.14}$$

where (i) comes from Lemma A.2, (ii) and (iii) come from (A.5), and (iii) and (iv) come from the definition of $\xi$ and $\sigma_k \le \sigma_1$. Since (C.14) ensures that (4.2) of Lemma 4.5, we obtain

$$\|\overline{V}^{(t+1)} - \overline{V}^{*(t+1)}\|_{\mathrm{F}} \le \frac{2}{\sigma_k}\|V^{(t+0.5)} - V^{*(t)}\|_{\mathrm{F}} \overset{(i)}{\le} \frac{2\sqrt{\delta_{2k}}}{\sigma_k}\|V^{(t)} - V^{*(t)}\|_{\mathrm{F}} + \frac{4}{\xi}\|\overline{U}^{(t)} - \overline{U}^{*(t)}\|_{\mathrm{F}}$$

$$\overset{(ii)}{\le} \left(\frac{\sigma_k}{3\xi\sigma_1} + \frac{4}{\xi}\right) \cdot \frac{\sigma_k^2}{4\xi\sigma_1^2} \overset{(iii)}{\le} \frac{\sigma_k^2}{4\xi\sigma_1^2}, \tag{C.15}$$

where (i) and (ii) come from (C.14), and (iii) comes from the definition of $\xi$ and $\sigma_1 > \sigma_k$. Moreover, since (C.14) ensures that (A.3) of Lemma A.3 holds, we have

$$\|U^{(t)} - U^{*(t+1)}\|_{\mathrm{F}} \le \frac{3\sigma_1}{\sigma_k}\|V^{(t+0.5)} - V^{*(t)}\|_{\mathrm{F}} + \sigma_1\|\overline{U}^{(t)} - \overline{U}^{*(t)}\|_{\mathrm{F}}$$

$$\overset{(i)}{\le} \frac{3\sigma_1\sqrt{\delta_{2k}}}{\sigma_k}\|V^{(t)} - V^{*(t)}\|_{\mathrm{F}} + \left(\frac{6}{\xi} + 1\right)\sigma_1\|\overline{U}^{(t)} - \overline{U}^{*(t)}\|_{\mathrm{F}}$$

$$\overset{(ii)}{\le} \frac{3\sigma_1}{\sigma_k} \cdot \frac{\sigma_k^3}{12\xi\sigma_1^3} \cdot \frac{\sigma_k^2}{2\xi\sigma_1} + \left(\frac{6}{\xi} + 1\right) \cdot \frac{\sigma_k^2}{4\xi\sigma_1}$$

$$= \left(\frac{\sigma_k^2}{4\xi^2\sigma_1^2} + \frac{3}{\xi} + 1/2\cdot\right)\frac{\sigma_k^2}{2\xi\sigma_1} \overset{(iii)}{\le} \frac{\sigma_k^2}{2\xi\sigma_1},$$

where (i) comes from (C.14), (ii) comes from (A.5), and (iii) comes from the definition of $\xi$ and $\sigma_1 \ge \sigma_k$. $\qquad\square$

## D   Algorithms for Matrix Completion

---

**Algorithm 2** A family of nonconvex optimization algorithms for matrix completion. The incoherence factorization algorithm $\mathsf{IF}(\cdot)$ is illustrated in Algorithm 3, and the partition algorithm $\mathsf{Partition}(\cdot)$, which is proposed by [10], is provided in Algorithm 4. The initialization procedures $\mathsf{INT}_{\overline{U}}(\cdot)$ and $\mathsf{INT}_{\overline{U}}(\cdot)$ are provided in Algorithm 5 and Algorithm 6. Here $\mathcal{F}_{\mathcal{W}}(\cdot)$ is defined in (5.2).

---

**Input**: $\mathcal{P}_{\mathcal{W}}(M^*)$
**Parameter**: Step size $\eta$, Total number of iterations $T$
$(\{\mathcal{W}_t\}_{t=0}^{2T}, \widetilde{\rho}) \leftarrow \mathsf{Partition}(\mathcal{W})$, $\mathcal{P}_{\mathcal{W}_0}(\widetilde{M}) \leftarrow \mathcal{P}_{\mathcal{W}_0}(M^*)$, and $\widetilde{M}_{ij} \leftarrow 0$ for all $(i,j) \notin \mathcal{W}_0$
$(\overline{U}^{(0)}, V^{(0)}) \leftarrow \mathsf{INT}_{\overline{U}}(\widetilde{M})$, $(\overline{V}^{(0)}, U^{(0)}) \leftarrow \mathsf{INT}_{\overline{V}}(\widetilde{M})$
**For:** $t = 0, ...., T-1$

   Alternating Exact Minimization : $V^{(t+0.5)} \leftarrow \mathrm{argmin}_V \mathcal{F}_{\mathcal{W}_{2t+1}}(\overline{U}^{(t)}, V)$ ⎫
   $(\overline{V}^{(t+1)}, R_{\overline{V}}^{(t+0.5)}) \leftarrow \mathsf{IF}(V^{(t+0.5)})$                                          ⎪
   Alternating Gradient Descent : $V^{(t+0.5)} \leftarrow V^{(t)} - \eta\nabla_V\mathcal{F}_{\mathcal{W}_{2t+1}}(\overline{U}^{(t)}, V^{(t)})$  ⎪
   $(\overline{V}^{(t+1)}, R_{\overline{V}}^{(t+0.5)}) \leftarrow \mathsf{IF}(V^{(t+0.5)})$, $U^{(t)} \leftarrow \overline{U}^{(t)} R_{\overline{V}}^{(t+0.5)\top}$  ⎬ Updating $V$.
   Gradient Descent : $V^{(t+0.5)} \leftarrow V^{(t)} - \eta\nabla_V\mathcal{F}_{\mathcal{W}_{2t+1}}(\overline{U}^{(t)}, V^{(t)})$         ⎪
   $(\overline{V}^{(t+1)}, R_{\overline{V}}^{(t+0.5)}) \leftarrow \mathsf{IF}(V^{(t+0.5)})$, $U^{(t+1)} \leftarrow \overline{U}^{(t)} R_{\overline{V}}^{(t+0.5)\top}$  ⎭

   Alternating Exact Minimization : $U^{(t+0.5)} \leftarrow \mathrm{argmin}_U \mathcal{F}_{\mathcal{W}_{2t+2}}(U, \overline{V}^{(t+1)})$ ⎫
   $(\overline{U}^{(t+1)}, R_{\overline{U}}^{(t+0.5)}) \leftarrow \mathsf{IF}(U^{(t+0.5)})$                                          ⎪
   Alternating Gradient Descent : $U^{(t+0.5)} \leftarrow U^{(t)} - \eta\nabla_U\mathcal{F}_{\mathcal{W}_{2t+2}}(U^{(t)}, \overline{V}^{(t+1)})$  ⎪
   $(\overline{U}^{(t+1)}, R_{\overline{U}}^{(t+0.5)}) \leftarrow \mathsf{IF}(U^{(t+0.5)})$, $V^{(t+1)} \leftarrow \overline{V}^{(t+1)} R_{\overline{U}}^{(t+0.5)\top}$  ⎬ Updating $U$.
   Gradient Descent : $U^{(t+0.5)} \leftarrow U^{(t)} - \eta\nabla_U\mathcal{F}_{\mathcal{W}_{2t+2}}(U^{(t)}, \overline{V}^{(t)})$       ⎪
   $(\overline{U}^{(t+1)}, R_{\overline{U}}^{(t+0.5)}) \leftarrow \mathsf{IF}(U^{(t+0.5)})$, $V^{(t+1)} \leftarrow \overline{V}^{(t)} R_{\overline{U}}^{(t+0.5)\top}$  ⎭

**End for**
**Output:** $M^{(T)} \leftarrow U^{(T-0.5)}\overline{V}^{(T)\top}$ or $\overline{U}^{(T)} V^{(T)\top}$ (Gradient Descent Only)

---

**Algorithm 3** The incoherence factorization algorithm for matrix completion. It guarantees that the solutions satisfy the incoherence condition throughout all iterations.

---

**Input**: $W^{\text{in}}$
$r \leftarrow$ Number of rows of $W^{\text{in}}$
**Parameter**: Incoherence parameter $\mu$
$(\overline{W}^{\text{in}}, R_{\overline{W}}^{\text{in}}) \leftarrow \mathsf{QR}(W^{\text{in}})$
$\widetilde{W} \leftarrow \underset{W}{\arg\min} \|W - \overline{W}^{\text{in}}\|_{\text{F}}^2$  subject to  $\underset{j}{\max} \|W_{j*}\|_2 \leq \mu\sqrt{k/r}$
$(\overline{W}^{\text{out}}, R_{\overline{W}}^{\text{tmp}}) \leftarrow \mathsf{QR}(W^{\text{out}})$
$R_{\overline{W}}^{\text{out}} = \overline{W}^{\text{out}\top} W^{\text{in}}$
**Output:** $\overline{W}^{\text{out}}, R_{\overline{W}}^{\text{out}}$

---

**Algorithm 4** The observation set partition algorithm for matrix completion. It guarantees the independence among all $2T + 1$ output observation sets.

---

**Input**: $\mathcal{W}, \bar{\rho}$
$\widetilde{\rho} = 1 - (1 - \bar{\rho})^{\frac{1}{2T+1}}$.
**For:** $t = 0, \dots, 2T$
$\quad \widetilde{\rho}_t = \dfrac{(mn)!\bar{\rho}^{t+1}(1 - \bar{\rho})^{mn-t-1}}{\bar{\rho}(mn - t - 1)!(t + 1)!}$
**End for**
$\quad \mathcal{W}_0 = \emptyset, \dots, \mathcal{W}_{2T} = \emptyset$
**For** every $(i, j) \in \mathcal{W}$
$\quad$ Sample $t$ from $\{0, \dots, 2T\}$ with probability $\{\widetilde{\rho}_0, \dots, \widetilde{\rho}_{2T}\}$
$\quad$ Sample (w/o replacement) a set $\mathcal{B}$ such that $|\mathcal{B}| = t$ from $\{0, \dots, 2T\}$ with equal probability
$\quad$ Add $(i, j)$ to $\mathcal{W}_\ell$ for all $\ell \in \mathcal{B}$
**End for**
**Output:** $\{\mathcal{W}_t\}_{t=0}^{2T}, \widetilde{\rho}$

---

**Algorithm 5** The initialization procedure $\mathsf{INT}_{\overline{U}}(\cdot)$ for matrix completion. It guarantees that the initial solutions satisfy the incoherence condition throughout all iterations.

---

**Input**: $\widetilde{M}$
**Parameter**: Incoherence parameter $\mu$
$(\widetilde{U}, \widetilde{D}, \widetilde{V}) \leftarrow \mathsf{KSVD}(\widetilde{M})$
$\widehat{U}^{\text{tmp}} \leftarrow \underset{U}{\arg\min} \|U - \hat{U}\|_{\text{F}}^2$  subject to  $\underset{i}{\max} \|U_{i*}\|_2 \leq \mu\sqrt{k/m}$
$(\overline{U}^{\text{out}}, R_{\overline{U}}^{\text{out}}) \leftarrow \mathsf{QR}(\widetilde{U}^{\text{tmp}})$
$\widetilde{V}^{\text{tmp}} \leftarrow \underset{V}{\arg\min} \|V - \widetilde{V}^{\text{tmp}}\|_{\text{F}}^2$  subject to  $\underset{j}{\max} \|V_{j*}\|_2 \leq \mu\sqrt{k/n}$
$(\overline{V}^{\text{out}}, R_{\overline{V}}^{\text{out}}) \leftarrow \mathsf{QR}(\widetilde{V}^{\text{tmp}})$
$V^{\text{out}} = \overline{V}^{\text{out}}(\overline{U}^{\text{out}\top} \widetilde{M} \overline{V}^{\text{out}})^\top$
**Output:** $\overline{U}^{\text{out}}, V^{\text{out}}$

---

**Algorithm 6** The initialization procedure $\mathsf{INT}_{\overline{V}}(\cdot)$ for matrix completion. It guarantees that the initial solutions satisfy the incoherence condition throughout all iterations.

---

**Input**: $\widetilde{M}$

**Parameter**: Incoherence parameter $\mu$

$(\widetilde{U}, \widetilde{D}, \widetilde{V}) \leftarrow \mathsf{KSVD}(\widetilde{M})$

$\widetilde{V}^{\mathrm{tmp}} \leftarrow \underset{V}{\operatorname{argmin}} \|V - \widetilde{V}\|_{\mathrm{F}}^2$ subject to $\underset{j}{\max} \|V_{j*}\|_2 \leq \mu\sqrt{k/n}$

$(\overline{V}^{\mathrm{out}}, R_{\overline{V}}^{\mathrm{out}}) \leftarrow \mathsf{QR}(\widetilde{V}^{\mathrm{tmp}})$

$\widetilde{U}^{\mathrm{tmp}} \leftarrow \underset{U}{\operatorname{argmin}} \|U - \widetilde{U}^{\mathrm{tmp}}\|_{\mathrm{F}}^2$ subject to $\underset{i}{\max} \|U_{i*}\|_2 \leq \mu\sqrt{k/m}$

$(\overline{U}^{\mathrm{out}}, R_{\overline{U}}^{\mathrm{out}}) \leftarrow \mathsf{QR}(\widetilde{U}^{\mathrm{tmp}})$

$U^{\mathrm{out}} = \overline{U}^{\mathrm{out}}(\overline{U}^{\mathrm{out}\top}\widetilde{M}\overline{V}^{\mathrm{out}})$

**Output:** $\overline{V}^{\mathrm{out}}, U^{\mathrm{out}}$

---