[Reviews · NeurIPS 2015]

Submitted by Assigned_Reviewer_1

The paper addresses convergence of alternating algorithms in non-convex formulations of matrix completion and matrix sensing. Global convergence with geometric rate is established under certain conditions.

The paper is very ambitious. It addresses 3 algorithms (alternating exact minimisation, alternating gradient descent, gradient descent) for 2 problems (completion and sensing). Because it is impossible to address the 6 all possible cases, many parts are either sketched or relegated to supplementary materials. As the results, the paper feels a bit scattered and doesn't read very well.

I would prefer that the authors address one special case rigorously and then explain how the results apply to other cases with minor modifications.

The experiments are trivial - they should be either discarded or greatly improved (preferred).

** General comments - the role of initialisation in the proposed convergence results is not clear. For example, it doesn't seem to intervene in the proposed Theorems (though this is announced in line 38). I suppose that initialisation is critical to obtain the global convergence ?

- after reading Section 3.1, it feels like the theoretical study is done, that Eq. on line 223 provides the necessary results. What is it that Theorems 3.4 and 3.6 specifically address in the context of the discussion of Section 3.1 ? - more generally, the paper doesn't sufficiently emphasises what are the difficulties that are being solved and doesn't provide enough intuitions about the solutions. For example, Section 4 is not very inspiring and does not sufficiently explains how the proof works at the general level. A key contribution seems to be the use of the projected Oracle divergence as defined in Eq. (3.5) and I wish it was more motivated. - how realistic/stringent are the conditions of Theorems 3.4 and 3.6 in practice ? - please explain the difference between alternating gradient descent and gradient descent - examining Algorithm 1 doesn't tell much. - the issue of identifiability and the chosen forms of U and V could be more discussed, as well as the relation between the respective convergences of U, V and M=UV.

** Specific comments - I don't see why the decomposition of M* in line 309 should depend on t ? - what's the meaning of Assumption 3.5 ? I don't see how it prevents sparsity. The use of star indices is confusing (clashes with the use of star to denote ground truth). - what's Omega in line 241, Eq. 3.10, etc ? - the sentence in lines 122-123 seems useless. - what's the renormalisation issue in line 188 ? - weird phrasing in lines 228-229. - caption of Fig.1 essentially repeats the text of Section 5.

** typos - entires - board - takeS line 75 - existS line 276 - noting that line 125 - that our line 253
Summary: The paper is very ambitious. It addresses 3 algorithms (alternating exact minimisation, alternating gradient descent, gradient descent) for 2 problems (completion and sensing). Because it is impossible to address the 6 all possible cases, many parts are either sketched or relegated to supplementary materials. As the results, the paper feels a bit scattered and doesn't read very well.

Submitted by Assigned_Reviewer_2

This paper studies nonconvex optimization algorithms for estimation of low rank matrices that include alternating minimization and gradient-type methods. It shows the geometric convergence to the global optimum for the algorithms and then the exact recovery of the true low rank matrices under standard conditions.

In terms of matrix completion (not matrix sensing), their result is not better than the result of Hastie et al. (2014) (http://arxiv.org/pdf/1410.2596v1.pdf) which they didn't cite in their paper. Please see Theorem 3, Theorem 4 plus the following statements, and Theorem 6 in Hastie et al..

You will see that Hastie et al. prove the same results as Theorem 3.6 of the given paper under milder conditions. However, since Hastie et al. didn't handle the matrix sensing problem, we might find the novelty of the given paper there.
Summary: Significant overlap with "Matrix Completion and Low-Rank SVD via Fast Alternating Least Squares" by Hastie, Mazumder, Lee, Zadeh.

http://arxiv.org/pdf/1410.2596v1.pdf

Submitted by Assigned_Reviewer_3

The paper extends the alternating minimization algorithm (reference [12]) to solve the low rank matrix estimation problem with general loss function. The assumptions for the general loss functions and the target matrix are standard. Though the extension is straight forward, the theoretical result for such general loss functions is meaningful. Please use matrix completion examples in the experiments, as matrix completion is more interesting and matrix sensing with RIP condition has been well investigated.
Summary: The paper extends the alternating minimization algorithm to solve the low rank matrix estimation problem with general loss function. Though the extension is straight forward, the theoretical result obtained in this paper is meaningful.

Submitted by Assigned_Reviewer_4

This paper studies low rank matrices via non-convex optimization. It looks at a variety of algorithms(alternating minimization, grad descent, etc.) and models (Gaussian, matrix completion). My worry however is that compared to the existing literature (Jain et. al., Lafferty et. al. ,Balakrishnan et.al., Candes. et. al., Montanari et. al.) does not improve things. Also the results are worse than convex schemes in terms of the dependence of the sample complexity on the rank r.

A few minor comments based on a quick read:

- page 1 line 53

The claim for matrix completion is not really correct as there is a dependence on $\epsilon$ in the number of measurements e.g. this result is worse than the result of Keshavan, Montanari and Oh

- page 3

Is the QR factorization necessary for the algorithm to work or just for the proofs to work? Some discussion about the complexity of QR method and its advantages help.

- Page 4 section 3.1 and remark 3.2 "2-norm of difference between gradients and to quantify the difference between the finite sample and population settings"

similar conditions to Projected oracle divergence has been proposed in the context of phase retrieval for analyzing non-cvx objectives which is not mentioned here

- Page 5, Theorem 3.4.

k^3 n log n. The results of Zheng and Lafferty already seems sharper but perhaps this was after NIPS

- Page 5, Remark 3.2

Viewing non-convex optimization as sort of an approximate gradient scheme and in particular the recommendation that ell_2 norm of gradient be used as means of analyzing gradient descent was proposed prior to [1] e.g. see the work of Candes et. al. on phase retrieval Sections 2.3 and 7.9

- Page 6 line 284

The dependence on the condition number maybe worse than [7]

Summary: This paper looks at the well studied problem of matrix sensing and completion using non-convex schemes. There are a few papers that already address this problem. This papers looks at a variety of gradient descent, alternating minimization, etc. Looking at the paper quickly it seems mostly sound. My worry however is that compared to the existing literature (Jain et. al., Lafferty et. al. ,Balakrishnan et.al., Candes. et. al., Montanari et. al.) the results are not sharper (e.g. the sample complexity is not improved w.r.t. existing literature and sometimes even worse). Also the assumption that the condition number is bounded is rather strong.

Submitted by Assigned_Reviewer_5

This paper looks at non-convex optimization approaches common to matrix sensing and completion, and have been found to be more scalable and accurate than the convex versions. The authors unify three types of non-convex approaches, provide theoretical results that demonstrate fast convergence and sample complexity for all these methods, and perform preliminary experiments on synthetic data.

I found the paper to be very well-written and clear, and was in fact a pleasure to read. I find the results to be quite strong, and overall quite satisfied with the paper. The proofs are quite difficult to read, and the appendix is unreasonably lengthy (I could not go through them in detail), so the authors should consider alternate presentation. For example, I would present one application in detail (say matrix completion), and leave all description of matrix sensing to the appendix.

- There is no question that this is a very important problem, with active research both on theoretical and practical side. The strong theoretical results are thus likely to make a considerable impact on the area.

- The authors removed the space reserved for author names from the draft, which is against the prescribed format. I don't think this is a major concern, but I definitely don't appreciate it.

- Although not introduced here, I find oracle divergence an intuitive idea, and like its application to this setting.

- A more direct comparison with convex versions would have been good. How do the theoretical convergence results compare to convex approaches? How do the convex approaches perform in their benchmarks?

- Given that the focus of this paper is theoretical, and I understand the space was limited as it is, I really appreciate the few experiments that the authors included to demonstrate the linear convergence and dependency on number of samples.

Summary: I think this is a very strong, we;;-written paper. The task being considered is a very important one, and the theoretical results are quite strong, providing new insights along with generalizing existing results. Length is an issue though, so I hope authors will be able to express the proofs more succinctly.

Submitted by Assigned_Reviewer_6

Minor comments ------------------- line 202: for any \eta, do you get this update? if not, say "for some \eta" figure 1: legend appears to be a duplicate of the text

line 415: "both algorithms fail BECAUSE (...) is below the minimum requirement": this is vague and misleading -- the experiments provided do not allow to draw this conclusion I believe, as all the bounds and results have leading constants != 1, and looking at three values of d is not enough
Summary: The paper performs an ERM-like analysis for some alternate optimization schemes for low-rank matrix estimation. Results are probabilistic, and correspond to common generative models (matrix sensing with random design, iid support matrix completion).

The main contribution is to show how state-of-the art results for these two models can be derived in a unified framework.

Author Feedback
Author rebuttal: We thank the reviewers for their valuable suggestions.

Reviewer 1:

1) We will focus on one special case and then explain how the results apply to other cases with modifications.

2) We will add more numerical examples under various settings to verify the tightness of the sample complexity.

3) The initialization is critical to the global convergence. We will emphasize its importance.

4) Sec.3.1 only presents the high level idea of our analysis. Theorems 3.4 and 3.6 further verify several technical conditions used in Sec.3.1 (bi-convexity, bi-smoothness, upper bounding the projected oracle divergence), and establish the rigorous convergence rates for specific examples

5) As emphasized in Sec.1, the difficulty of the convergence analysis for nonconvex matrix factorization has been widely recognized in existing literature. There was no significant breakthrough until recent discovery in [12]. Section 4 contains the detailed proof. The intuition behind our proof and the motivation of the projected oracle divergence are presented in Sec.3.1

6) The technical conditions in both theorems are almost necessary from the information-theoretic perspective but not stringent in practice. They can be guaranteed by a sufficiently large sample size. Moreover, these conditions are very common, and have appeared in all existing literature on nonconvex matrix factorization e.g. [7-12]

7) The difference between gradient descent and alternating gradient descent: At the t-th iteration, the gradient descent uses the gradient w.r.t V and U at U=U^t and V=V^t to decrease the objective. But the alternating gradient first uses the partial gradient w.r.t V at U=U^t and V=V^t, and then uses the partial gradient w.r.t. U at U=U^t and V=V^{t+1} to decrease the objective

8) As explained in Remark 4.2, the matrix factorization is not unique (up to rotation). To resolve the ambiguity, we show that there always exits a decomposition of M^* such that the solution U^t and V^t at each iteration satisfy the desired optimization error bound. This is why we consider a "reference" decomposition of M^* for each t. It is worth noting that this is only an intermediate step for our analysis, since the final goal is to analyze the reconstruction error of M^*.

9) We will change the claim to "Under Assumption 3.5, each entry of M^* is expected to provide the similar amount of information". Similar claims can be found in [2,12,13,22]

10) a_n = Omega(b_n) means there exists some constants C and N such that a_n>=C*b_n for all n>N

11) The renormalization issue: We want to preserve the orthonormality of U^t and V^t. Thus we use QR to renormalize U^t and V^t to orthonormal matrices

Reviewer 2:

1) Our extensions are not only on the loss function, but more importantly on the types of algorithms. Beyond alternating minimization, we establish the convergence rate for gradient-type methods, which has not been established before. Our theory builds on the notion of projected oracle divergence instead of the perturbed power method in [12], and is thus not straightforward

2) We will use matrix completion for the numerical example, and move matrix sensing to the appendix

Reviewer 3: We will add more numerical examples under various settings to verify the tightness of the sample complexity, and compare the nonconvex methods with existing convex methods

Reviewer 4: We will add Hastie 2014 to the reference. It is worth noting that Hastie 2014 establish a sublinear convergence to local optima, which coincides with the global optima under additional conditions. But we establish a linear convergence to the global optima directly

Reviewer 5:

1) Our paper aims to provide a general and simple framework for analyzing nonconvex matrix factorization. The sample complexity is sightly sacrificed to gain maximum generality of our analysis. In fact, the sample complexity and dependency on condition numbers can be further improved if our analysis is specifically geared towards certain algorithms

2) We will change the claim in Sec.1 to "For matrix completion, the algorithm can recover the true matrix up to an \eps accuracy depending on the sample size". However, it is worth noting that the dependence on \eps arises from the analysis for the convergence rate, while Keshavan, Montanari and Oh only establishes the asymptotic convergence

3) QR is not necessary for alternating exact minimization in both theory and practice. For other algorithms, currently we need QR for their convergence analysis, though these algorithms actually work well without QR in practice. We will also add additional discussions

4) Thanks for pointing out the relevant paper on analyzing noncvx optimization in phase retrieval. We will add the reference and additional discussions

Reviewer 6: We will fix the minor issues and add more numerical experiments under different settings (varying d,k,m,n) to further verify the tightness of the sample complexity.